# A 3D Geological Model as a Base for the Development of a Conceptual Groundwater Scheme in the Area of the Colosseum (Rome, Italy)

**Cristina Di Salvo** [1],*, **Marco Mancini** [1], **Gian Paolo Cavinato** [1], **Massimiliano Moscatelli** [1], **Maurizio Simionato** [1], **Francesco Stigliano** [1], **Rossella Rea** [2] **and Antonio Rodi** [3]

[1] CNR-Institute of Environmental Geology and Geoengineering, Area della Ricerca di Roma 1—Montelibretti Via Salaria km 29,300, 00015 Monterotondo (RM), Italy; marco.mancini@igag.cnr.it (M.M.); gianpaolo.cavinato@igag.cnr.it (G.P.C.); massimiliano.moscatell@igag.cnr.it (M.M.); maurizio.simionato@igag.cnr.it (M.S.); francesco.stigliano@igag.cnr.it (F.S.)

[2] Soprintendenza Speciale per il Colosseo, Museo Nazionale Romano e l'Area Archeologica di Roma (Currently the Parco Archeologico del Colosseo), Piazza S. Maria Nova, 53, 00186 Roma, Italy; rossella.rea@beniculturali.it

[3] Geoter s.r.l., Viale Piramide Cestia, 31, 00153 Roma, Italy; rodi.ant@libero.it

* Correspondence: cristina.disalvo@igag.cnr.it

**Abstract:** Geological models are very useful tools for developing conceptual schemes owing to their capacity to optimize the management of stratigraphic information. This is particularly true in areas where archaeological heritage is exposed to hydrogeological hazards; 3D models can constitute the first step toward the construction of numerical models created to understand processes and plan mitigation actions to improve visitor safety and preserve archaeological heritage. This paper illustrates the results of a 3D hydrostratigraphic model of the site of the Colosseum in the Central Archaeological Area of Rome. In recent years, this area has experienced numerous floods caused by intense meteorological events. A new borehole survey provided the opportunity to update previous maps and cross sections and build a local scale 3D model. The resulting conceptual model was used to identify primary gaps in existing knowledge about the groundwater system and to optimize the planning of a piezometer monitoring network. Further studies can then focus on the development of groundwater numerical models to verify hypotheses regarding inflow-outflow dynamics and facilitate the optimization of water management.

**Keywords:** 3D geological models; groundwater conceptual model; Colosseum; hydrogeological hazards

---

## 1. Introduction

A geological model can be considered a three-dimensional (3D) spatial representation of the distribution of sediments and rocks below the ground surface. Traditionally, the results of geological and geophysical data collection are presented in 2D geological maps and cross sections; this methodology can be considered adequate only when the subsurface effects of the 3D structure are not relevant for solving a particular problem. Otherwise, a 3D representation of subsurface geometry, with associated hydrogeological, geotechnical and geophysical characteristics, is required.

A geological model aids the interpretation of the geometry, thickness and variable spacing of the geological units that control subsoil and surface fluid movements and the mechanical response of the ground to building loads, seismic action and groundwater-induced settlements. Specifically, a complex setting of subsurface layers can imply relevant spatial variations to texture, cohesion and geotechnical

characteristics, as well as hydraulic conductivity values. Depending on the direction of the hydraulic gradient and piezometric head, spatial relationships between geological formations and different facies can achieve either the compartmentalization of the flow regime or the flow exchange between units. They also determine the conditions of aquifer confinement or communication, groundwater storage and release rates. Developing a 3D geological model brings many advantages. Interpreted geology can be compared to field and subsurface data from direct and geophysical approaches in a 3D environment. After each adjustment and refinement, the model is run again and all sections and 3D geology are automatically updated, incorporating all previous improvements and avoiding unrealistic interpretations. A detailed 3D geological model can also improve the capability to understand the hydrogeological hazards to which an urban area is generally exposed, as extreme events associated with water occurrence, movement and distribution ([1–4]). This application is particularly useful when dealing with ancient archaeological heritage and constitute a support to understand the potential impact of spatial-temporal hydrogeological variations on archaeological remains [5].

The case study presented here concerns the area of the Colosseum (Flavian Amphitheatre) in the historical center of Rome. Without a doubt one of the most significant legacies of the Roman Empire, it is considered one of the most famous monuments in the world. The Colosseum receives the highest annual number of visitors to all cultural heritage sites in Italy (6.5 M in 2016, [6]), captivating thousands of people with its magnificence. However, such a high flow of tourists has raised the exposure to hazards such as earthquakes and flooding and, consequently, generated an increase in overall risk. This poses the important challenge of improving safety for people while ensuring the optimal preservation of archaeological heritage. In recent years, the archaeological area in the center of Rome has experienced numerous floods caused by intense storm events ([7,8]). In particular, during a storm on 20 October 2011, the underground structures ("hypogea") of the Colosseum were quickly flooded by a copious amount of water that reached a depth of 6 m from the lowest level of the *hypogea*. The susceptibility of the study area to flooding is related to its hydrological setting. The Colosseum is situated in a topographical depression, where both surface water and groundwater converge. During the Republican and Imperial ages (510 BC–476 AD) lowlands such as stream valleys and floodplains, humid and subject to frequent flooding, served functioning archaic mills or as recreational areas or military training grounds. From the archaic age, a network of surficial channels and underground sewers (*cloacae* in Latin) was built to reclaim portions of terrains in Rome, which then become available for public activities, for example, the Campus Martius and the Circus Maximus [9]. With the decline of the Roman Empire, these drainage channels were largely abandoned, in some cases filled by flood-related sediment delivered by the Tiber River and its tributaries. Though partially replaced by modern sewers, the network appears insufficient in the event of intense storms.

Comprehending surface water-groundwater dynamics and interaction, together with anthropic modifications of the hydraulic system, is fundamental to understanding flood mechanisms and optimizing stormwater management. While many projects have focused on the geological setting of the area of the Colosseum [10–12], many uncertainties and unresolved issues remain with regards to the local groundwater system. A geological survey, comprising a borehole drilling and piezometric measurement campaign, was commissioned by the former Soprintendenza Speciale per il Colosseo, Museo Nazionale Romano e l'Area archeologica di Roma (currently the Parco Archeologico del Colosseo, https://parcocolosseo.it) to better understand the geological characteristics of the subsoil, including its hydrogeological features. The survey was carried out between May–July 2017 under the scientific supervision of the Institute of Environmental Geology and Geoengineering of CNR. In this text, the authors describe an updated and local scale hydrostratigraphic reconstruction of the area of the Colosseum, based on a 3D geological model incorporating this new log data.

Large scale geological and hydrogeological literature studies were acquired (e.g., References [13–16]). Since hydrogeological studies usually comprise the partitioning of the groundwater zone into units that provide the framework for describing groundwater flow ([17]), the geological units were distinguished based on the differences in granulometric composition, assuming that coarser sediments can be more

permeable; this means performing a distinction based on the hydrostratigrapy. Hydrostratigraphic complexes are defined as a geological domain classified in regard to its water-bearing characteristics [18], hydraulically consistent at a specified spatial scale, which must have a measurable thickness and aerial extent in order to be detected and, ideally, monitored. Following previous experiences of geological characterization in archaeological areas [13], the objective of this study is to furnish a robust base for further research focused on the mitigation of hydrogeological hazards. In this framework, the specific goal of the present study is to detect primary gaps in knowledge about the groundwater system in the area of the Colosseum, which should be filled in order to develop a numerical model designed to optimize groundwater management strategies. To achieve this result, the work proceeded according to the following steps—(1) revision of existing geological maps, principally in terms of formation boundaries and thicknesses; (2) evaluation of geometries and relationships between distinct hydrostratigraphic complexes; (3) definition of a groundwater conceptual model by detecting connections between recent alluvial valley deposits and surrounding older sedimentary sequences encasing this infill. A groundwater conceptual model (or scheme) consists of a series of hypotheses about how groundwater moves into the subsoil, made in order to understand the complexity of the groundwater system, which constitutes the first step toward the development of numerical models.

## 2. Geological and Hydrogeological Overview of the Study Area

The city of Rome is located approximately 25 km from the Tyrrhenian Sea coast and lays at the feet of the western central Apennines (Figure 1). The area is characterized by a stratigraphic sequence resulting primarily from Quaternary volcanism and glacio-eustatic changes of sea level that controlled the erosion and filling of fluvial paleo-valleys [19]. An up to 900 m thick mass of Pliocene marine sediments, which cover with angular unconformity Mesozoic–Cenozoic carbonate and siliciclastic sequences [20–22] are traditionally considered the hydrogeological bedrock of Rome [23–25]. The Pliocene sediments are in turn overlain by Lower-Middle Pleistocene shallow marine sediments and a volcano-sedimentary sequence from the late Early Pleistocene to the Late Pleistocene. This volcano-sedimentary sequence forms a plateau ranging in thickness from 30 to 150 m, composed of inter-layered fluvio-lacustrine sediments and distal pyroclastics and lavas from the nearby Monti Sabatini and Alban Hills volcanic complexes (References [26,27] with references). The valleys of the Tiber and its tributaries are deeply carved into the plateau sequence and filled with up to 65 m of fluvial sediments, referable to the Late Pleistocene-Holocene [28–31]. Lastly, the urban and archaeological areas are covered by a layer of anthropogenic deposits ranging in thickness from 1 to at least 20 m [32] and resulting from the human activity which deeply modified the previous topography, often filling the valleys. These works lead to inevitable changes to the original drainage and surface water flow increases the risk of flooding in the case of extreme rain events [33–35]. Following stratigraphic and depositional criteria, various authors have compiled lists of the typical geological sequences in the center of Rome [26,36–38], focusing on local geotechnical features [28,39], seismic properties [14,40,41] and hydrostratigraphic characteristics [24,42]. These models address the investigation of mechanisms inducing natural hazards such as settlements [43], flooding [8] and seismically induced amplification effects and permanent ground modifications [44–46]. In this work, the reference geological units, that is, synthems and formations are those detected in References [26,29]. Then, formations were grouped into hydrostratigraphic complexes, following the subdivisions into lithofacies and lithotypes proposed by Reference [14] (Table 1) for the nearby Palatino area (Figure 2A). The hydrostratigraphic complexes also take into account the given ranges of hydraulic conductivity values; the hydrogeological complexes identified in the reference works (Table 2) differ from the hydrostratigraphic complexes as regard to their behavior evident in field head monitoring.

The multiple aquifers recognized in the Rome area [10,24,47] can be ascribed to different hydrogeological units. A hydrogeological unit represents a group of hydrostratigraphic (or hydrogeological) complexes showing, at a regional scale, homogeneous hydrogeological characteristics (e.g., groundwater flowpaths, recharge areas) and have, generally, a uniform genetic

process. The delineation of hydrogeological units relies on both geologic and hydraulic data at a scale appropriate to the problem or question.

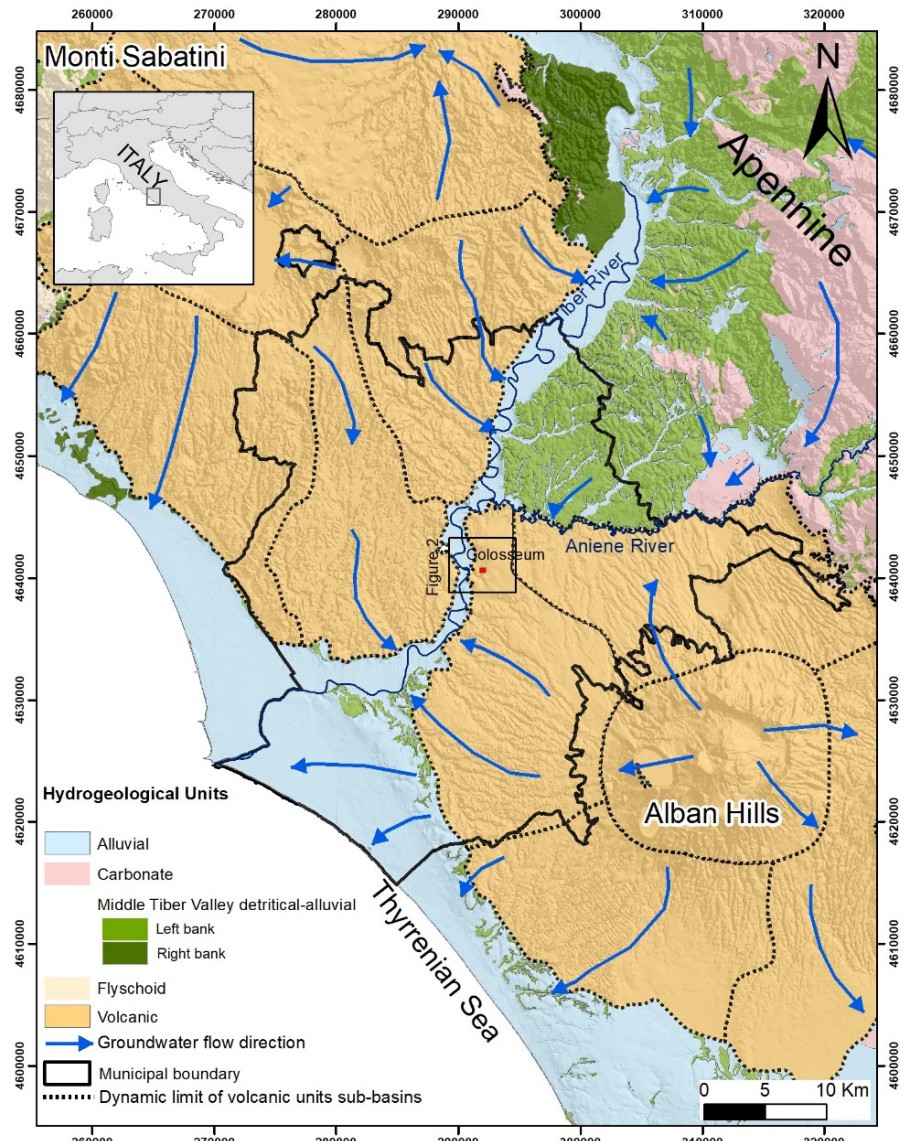

**Figure 1.** Regional Hydrogeological Scheme and Hydrogeological Units (From Reference [15], modified).

The Middle Pleistocene Detritic Alluvial Hydrogeological Unit (hereinafter, HU,) of the Middle Tiber Valley [15], correspond with the "Paleotiber" units of References [48,49], referred to as the Santa Cecilia Formation and the Fosso del Torrino Formation, respectively in more recent works by References [26,50] (Table 1 and Figure 1). It ranges in thickness from a few meters to more than 100 m in morpho-structural depressions and is characterized by a gravelly basal layer [22,26,51] overlaid by a prevalently clayey portion. The Detritic Alluvial HU hosts regional groundwater circulation, mainly directed in a NS direction; it is predominantly recharged by rainfall via the outcrops located along the Middle Tiber Valley. To the south (left bank of the Aniene River), this unit is covered by the thick Alban Hills volcanic HU.

The Central Archaeological Area of Rome (Figure 2A) lies in the core of the historical center, on the left bank of the Tiber River. This area comprises the city's historically renowned seven hills [52], namely the Quirinal, Viminal, Capitoline, Oppian-Esquiline, Palatine, Caelian and Aventine, which represent peripheral remnants of the volcano-sedimentary plateau at the NW margin of the Alban

Hills Volcanic District. The hills feature a mean elevation of approximately 40–50 m a.s.l. and are separated by narrow valleys, such as the Velabro, Murcia and Labicano, which once hosted tributaries of the Tiber River [11,37,53,54].

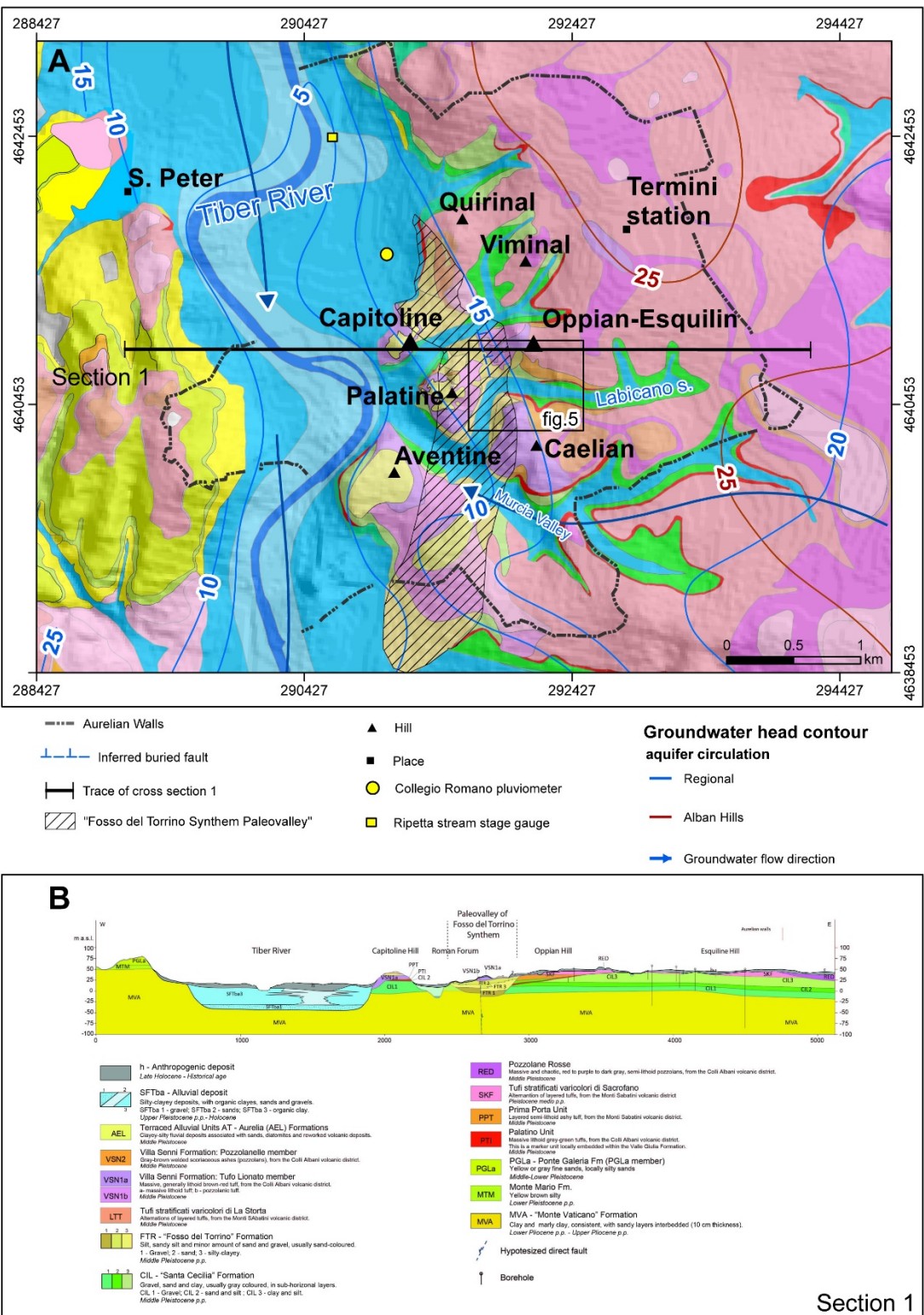

**Figure 2.** (**A**) Geological units (simplified from 26]) and groundwater head contour (after [55]). (**B**) Geological section 1 crossing the historical center (after [45]).

**Table 1.** Stratigraphic scheme reporting formal lithostratigraphic, synthemic and sequence stratigraphic units, from (1) [26] and (2) [29], broken down into component lithotypes, from Reference [14] (mod). Radiometric age dates (3) are from Reference [56] and references within. MIS indicates the Marine Isotope Stage of the Quaternary Period (see Reference [57] for further details). The last column on the right-hand side lists the hydrostratigraphic complexes used to group lithotypes for the purpose of this work. (*) Ponte Galeria and Monte Mario complexes are present in the right bank of Tiber River (see Figure 2A,B) and, therefore, are absent in the model's area. Related hydrogeological parameters and comparisons with literature hydrostratigraphic frames are in Table 2.

| Non Marine Synthems [1] and 3rd Order Marine Sequence [2] | Formation [1] | Lithotypes | Hydrostratigraphic Complex |
|---|---|---|---|
| Tiber River Synthem (MIS 5-1) | Anthropogenic layer (Holocene) | hm: dominant masonry<br>hb: dominant infill | h |
| | Alluvial deposit (Tiber River depositional system, Upper Pleistocene-Holocene) | SFTba 1: gravels<br>SFTba 2: sands<br>SFTba 3: silts and clay | SFTba 1<br>SFTba 2<br>SFTba 3 |
| Quartaccio Synthem (MIS 10-9) | Aurelia Fm.<br><br>Villa Senni Fm.-Pozzolanelle<br><br>Villa Senni Fm.-Tufo Lionato (355 ± 2 ka [3]) | AEL: sandy-silty<br>VSN2: poorly cemented, welded coarse scoriaceous ashes<br>VSN1 a: lithoid tuff<br>VSN1 b: poorly cemented, welded coarse scoriaceous ashes | VTA-Q |
| Fosso del Torrino Synthem (MIS 12-11) | Pozzolane nere Fm. (407 ± 4 ka [3]) + LTT<br><br>La Storta Fm. (416 ± 6 ka [3])<br><br>Pozzolane Rosse Fm. (457 ± 4 ka [3])<br><br>Fosso del Torrino Fm. | PNR: Black massive and chaotic pyroclastic unit<br>LTT: Ashy and scoriaceaous pyroclastic deposit<br>RED: massive semicoherent deposit with up to 24 cm diameter scoria | VTA-S |
| | | FTR 1: sandy gravels<br>FTR 2: silty sands and sandy silts<br>FTR 3: clayey silts and silty clays | FTR 1<br>FTR 2<br>FTR 3 |
| Villa Glori Synthem (MIS 14-13) | Tufi stratificati varicolori di Sacrofano Fm. (488 ± 2 ka [3])<br><br>Prima Porta Unit (518 ± 5 ka [3])<br>Palatino Unit (520 ± 8 ka [3])<br>Valle Giulia Fm. | SKF: pyroclastic deposit with interbedded volcano-sedimentary layers.<br>PTI: lithoid tuff<br>PPT: lithoid tuff<br>VGU 1: gravels<br>VGU 2: silty sands, sandy silts and clays | PTI-VGU |
| Flaminia Synthem (MIS 16-15) | Santa Cecilia Fm. | CIL 1: gravels<br>CIL 2: interbedded silty sands and sandy silts<br>CIL 3: Clayey silt | CIL 1<br>CIL 2<br>CIL 3 |
| Magliana Synthem | Ponte Galeria Fm. | PGL$_a$: sands and silty sands | PGL * |
| Monte Mario Sequence | Monte Mario Fm. | MTM: sands and silts | MTM * |
| Vatican Sequence | Monte Vaticano Fm. (Lower–Upper Pliocene) | MVA | MVA |

Pleistocene sedimentary deposits crop out along the valleys carved out by streams, at the outer margin of the volcanic edifice. Groundwater flow hosted in volcanic complexes moves from the local piezometric high point (25 m a.s.l., corresponding to the Termini Railway station; Figure 2A) toward the draining stream valleys, where it merges with the circulation of the Pleistocene sedimentary complexes (i.e., the Detritic Alluvial HU). The groundwater base level is represented by the Tiber River, which acts as the discharge outlet under normal base flow conditions. (Figure 2B).

**Table 2.** Literature hydrogeological schemes (orange headed columns) compared with the hydrostratigraphic frame used in this work (light blue headed columns).

| Hydrogeological Unit (Capelli et al., 2012) | Hydrogeological Complex | | | Hydrostratigraphic Frame (Current Work) | k (m/d) [1] | Hydrogeological Condition in the Study Area |
|---|---|---|---|---|---|---|
| | Capelli et al., 2008 | Di Salvo et al., 2012 | La Vigna et al., 2016 | | | |
| - | Anthropic backfill C. | Anthropic backfill—Complex 5 | Anthropogenic deposit C. | h | - | Uppermost aquifer; variable transmissivity depending upon local thickness, granulometry and texture |
| Alluvial | Alluvium deposits C. | Recent alluvium—Complex 4 | Alluvial deposit C. | SFTba 3 | 0.00003–0.5 | aquiclude |
| | | | | SFTba 2 | 0.032–43.2 | mostly unconfined perched aquifers |
| | | | | SFTba 1 | 0.003–6.5 | confined aquifer, buried at the bottom of alluvial valleys |
| Volcanic | Complexes owing to the Alban Hills Hydrogeological Unit | Volcanic units | Heterogeneous clastic deposit C., "Tufo Lionato" C., High permeability Alban Volcanic deposit. | VTA | 0.025–5.7 [2] | Multilayer aquifer |
| | Pisolitic tuff and Sabatini Volcanic Complexes / Valle Giulia Formation Complex | Volcanic units / Coeval alluvial deposits / Complex 3 | Low permeability volcanic deposits of Alban Hills; Sabatini volcanic C. / Valle Giulia Formation Complex | PTI_VGU | 0.02–2.6 / 0.002–0.02 | mostly aquifer; k values depending on texture, alteration state, fracture density |
| Middle Tiber Valley detritical-alluvial | Fluvial-marshy complex of Santa Cecilia Fm C. | Ponte Galeria Unit-Complex 2 | Santa Cecilia Fm. C. (sandy silty portion) | FTR3 | 0.000397 | aquiclude |
| | | | | FTR2 | 0.14 | confined/unconfined aquifer |
| | | | Santa Cecilia Fm. C. (gravelly portion) | FTR1 | 0.292 | confined aquifer |
| | | | Santa Cecilia Fm. C. (sandy silty portion) | CIL3 | 0.0001–0.01 | aquiclude |
| | | | | CIL2 | | |
| | | | Santa Cecilia Fm. C. (gravelly portion) | CIL1 | 0.02–2 | confined/unconfined aquifer |
| | Gravels and sands of Ponte Galeria Fm C. | Ponte Galeria Unit-Complex 2 | Gravelly-Sandy "Ponte Galeria" Complex | PGLa | 0.01–0.3 | (3) |
| - | Coarse sands of Monte Mario and Ponte Galeria formations C. | Monte Mario Unit-Complex 1 | Coarse sands of Monte Mario and Ponte Galeria C. | MTM | 0.01 | (3) |
| - | Monte Vaticano clay C. | Mone Vaticano Unit—Complex 1 | Sandy-clayey basal C. | MVA | 0.0001–0.01 | regional basal aquiclude |

(1) Adapted from Mancini et alii, 2014

(2) due to the scarce extention of AEL Fm. in the study area, k values are those of Volcanic Units

(3) Complex not in the study area; appears in Figure 2

Four streams, now hidden by urbanization, flow toward the center of the modelled area of the Colosseum from the surroundings—Oppian-Esquiline hill, Caelian hill, Labicano Valley and the Via dei Fori Imperiali [58,59]. The area, centered around the Flavian Amphitheatre, encompasses a portion of a narrow and bending section of the 100 m wide and 2 km long Holocene Labicano stream valley. This valley runs in an EW direction, from its head toward the Colosseum, brusquely turning southward in correspondence with the Palatine hill footwall, before entering the valley of the Velabrum Maius (Vallis Murcia, corresponding with the Circus Maximus). Part of the foundations of the Colosseum are thus built Upper Pleistocene-Holocene sedimentary infill of the Labicano Valley. The bulk of the infilling alluvial sequence is represented by sandy-clayey sediments (SFTba 3 hydrostratigraphic complex; see Table 1) with a gravel layer at its base (SFTba 1 h. complex) and sandy lenses are sporadically interbedded in the sequence (SFTba 2 complex).

Shallow groundwater circulation occurs in the sandy portions of alluvium and the overlaying anthropogenic backfill deposits, affecting the first 7–8 m below the ground surface [60], recharged primarily by rainfall infiltration, aqueducts and sewer network losses. The granulometry and texture of anthropogenic backfill is very inhomogeneous, with a high variable hydraulic conductivity ranging from medium to high [32]; due to its variable thickness and permeability, it cannot be considered a confining layer and groundwater exchanges can occur with the underlying units. Deeper groundwater circulation occurs in the alluvial gravel basal layer (SFTba 1), showing different chemical characters with respect to both the shallower and deeper domains [10]. Regional groundwater circulation occurs in the Middle Pleistocene sedimentary formations (i.e., Santa Cecilia and Fosso del Torrino Fms). The Santa Cecilia Fm, hereinafter CIL Fm, constitutes a sub-horizontal, tabular and laterally continuous body extending over almost the entire eastern sector of the study area; to the west it is partially eroded by a 40 m deep and 0.5 km wide younger paleovalley incision running NS (Figure 2B); this incision represents the basal unconformity of the Fosso del Torrino synthem (represented here by the homonymous Fosso del Torrino Fm, hereinafter FTR Fm), carved directly into the Pliocene clayey substratum (Monte Vaticano Fm, MVA Fm). The CIL and FTR Fms are both characterized by an approximately 10 m thick coarse sands and gravel bed at their base, hosting highly permeable aquifers—the CIL 1 hydrostratigraphic complex, confined (where overlapped by the CIL 2 and CIL 3 complexes or by the clayey SFTba 3 complex) or semi-unconfined (when barely cropping out beneath the anthropogenic backfill deposit, h); the FTR 1 complex, confined by the overlapping silty-clayey FTR 3 complex and the silty sandy FTR 2 complex (Tables 1 and 2). The FTR 2 complex is a less transmissive aquifer with respect to FTR 1, confined (where overlapped by SFTba 3) or semi-unconfined (when barely cropping out beneath the anthropogenic backfill deposit, h). Finally, the PTI_VGU complex (made of alternated tuffs and silty sands), with variable permeability depending on locally different textures, alteration state and presence of fractures, hosts an unconfined aquifer.

Reconstructions of geological surfaces and calculations of lithology volumes are rare and often cover limited areas. The top of the MVA Fm has been mapped at the municipal scale [61,62]; sub-basin scale mapping of the overlaying Pleistocene sedimentary deposits also exists [24,42,55]. Larger-scale maps of the historical center represent the top of the MVA Fm, the bottom of volcanic deposits [62] and the bottom of the anthropogenic backfill deposits [32]. In the specific study area, a gravelly aquifer at the base of the Pleistocene sedimentary sequences was already documented [60]. However, a detailed description of local scale systems, useful to the investigation of groundwater relationships between complexes or to the construction of a conceptual model supporting numerical simulations, is currently lacking.

## 3. Hydrological Setting and Its Anthropogenic Modifications

Many authors have documented the hydrological peculiarities of this area. Gaius Suetonius Tranquillus [63–65] describes a *stagnum* (pool in Latin) dug in the area subsequently occupied by the Colosseum, located inside a large square artificial basin measuring 399,745 m$^2$ and between 4–6 m deep [66]. Prior to the digging of this pool, the area was reclaimed around the 6th century BC by means

of surface and underground channels. Geochemical surveys conducted by the ACEA-ATO2 Company in 2012 reveal ancient sewer networks running NS from the Colosseum area (corresponding with the location of the *Meta Sudans*, Figure 3A) toward the Valle Murcia-Circus Maximus along Via di San Gregorio. This system appears mainly to drain groundwater and only partially collects water losses from aqueducts (20% of total sewer discharge [67]).

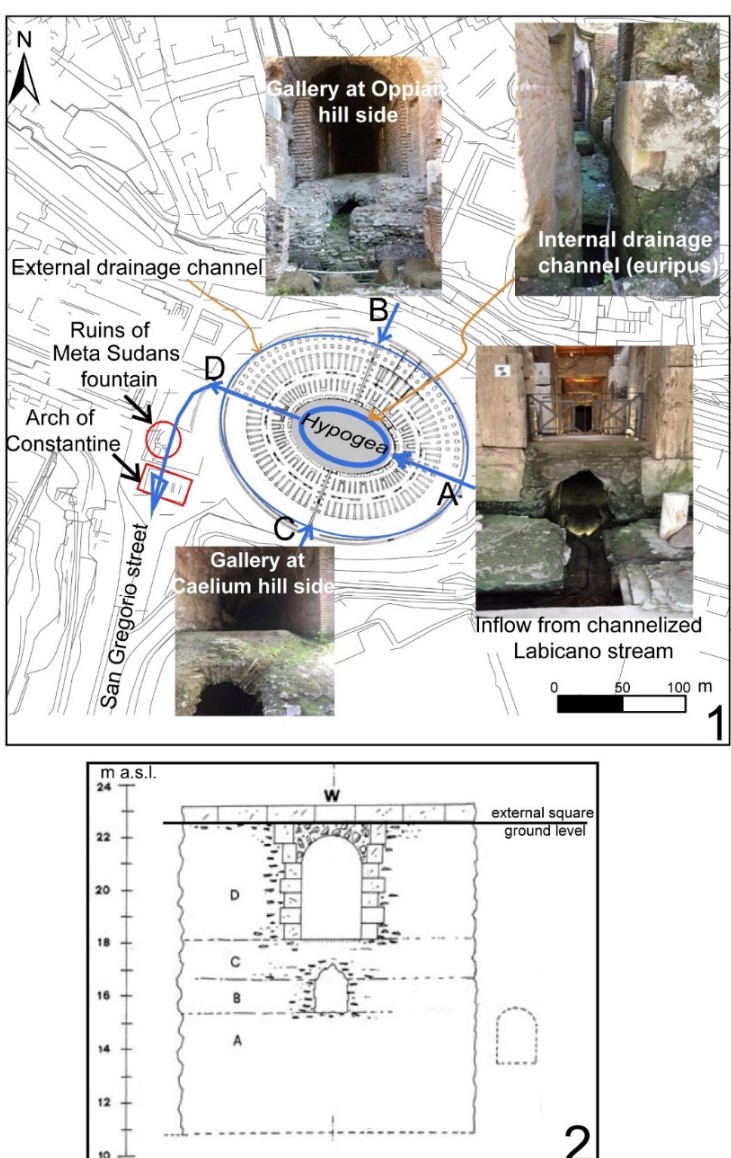

**Figure 3.** Sketch showing, in blue, the system of axial galleries and underlying drainage channels, radially crossing the upper foundation ring and of the annular channels, external and internal (*euripus*) to the foundation ring. (**1**) A: SE gallery and the inflowing Labicano stream channel; B: Oppian hill side tunnel; C: Caelian hill side tunnel; D: NW gallery and channel directed toward the external sewer (the outflowing Labicano stream). Note the abrupt, right angle deflection of flow of the Labicano stream to the south, immediately outside the Colosseum close to the *Meta Sudans* fountain. (**2**) Section showing the upper foundation ring with one of the galleries (D) and underlying channel (B) and, to the right, the deeper of the external sewer conduits.

The Colosseum was built between 70–80 AD by the Flavian Emperors. Lucius Cassius Dio ([65,68]) documented the practice of filling the Flavian Amphitheatre with water for aquatic spectacles (*naumachiae*, that is, mock ship battles) staged by the Emperors Titus and Domitian. It is documented

that a branch of the Claudio Aqueduct supplied water to the Caelian Hill and its surroundings [69]; water was diverted into *cisternae* (cisterns; [70]) and then delivered toward the Colosseum, further north, by channels [71]. The hydraulic head of the aqueduct was high sufficient to distribute water to fountains and hygienic services at the third level of the Colosseum [72]. Reference [73] used an engineering analysis to quantify the aqueduct water supply required to fill the *hypogea* below the Colosseum, estimating the time for water filling during recreational events and subsequent restoration. However, hypotheses about the system supplying water to the area of the Arena are still largely unexplored.

The foundations of the Colosseum consist of two overlapping concrete rings [70] hosting an impressive system of galleries and sewers. Sewers are embedded in the lower foundation, consisting in two systems of annular conduits running externally to the monument (Figure 3.2). The shallower conduit is 60 cm wide and 1.6 m high and situated 2.80 m below the square in front of the Colosseum [72]; the deeper and largest conduit, measuring $1.30 \times 1.80$ m (width/height), lies at a depth of about 8 m below the external square and is connected to the shallower conduit by wells positioned at regular intervals. While the shallower conduit was used to collect rainfall, the dimension of the deepest conduit suggests the ability to collect and drain notable volumes of water, probably not only due to rainfall runoff and supply water in excess of what was required [74].

The upper foundation, in turn, is a thick elliptical ring surrounding the *hypogea* area, ranging in depth between 22 m a.s.l. (i.e., the ground elevation of the external square) and 16 m a.s.l. (i.e., the ground elevation of the hypogea). The ring is crossed by four radial galleries, with underlying channels, aligned along the axes of the ellipse (A, B, C, D in Figure 3.1).

A perennial stream can be observed from Gallery A toward the hypogea, below the *Porta Libitinaria* gate, substantially due to the channeling of the Labicano stream, realized in conjunction with the reclamation of the area [70,75]. The actual discharge of the Labicano, corresponds to an average of 1 l/s [76]; actually, no data exist about changes of meteoclimatic conditions, in such a way as to suggest a significant variation of the historical discharge. However, even a three times higher discharge would be insufficient to justify the dimension of the external sewer conduit. Galleries B and C connect the hypogea to the base of Oppian and Caelian hills, respectively. Recent hydrogeological studies and speleological surveys revealed an open reservoir below the Caelian Hill [77,78] probably collecting groundwater emerging from the sedimentary aquifer (i.e., the CIL 1 Complex, confined between the bedrock, MVA Fm and the overlaying volcanic multilayer). This information permits the supposition that retained groundwater could inflow from Gallery C (Caelian hill, south side), in addition to the documented aqueduct water supply. Very little is known about Gallery B (Oppian Hill side); however, due to its hydrogeological setting, similar to the one at the Caelian Hill, it is not possible to exclude that it served to bring groundwater toward the hypogea from the north. This hypothesis is supported by the observation of water percolating from Gallery B at the level of the hypogea during the 2017 survey. Galleries A, B and C are connected to the elliptic channel (the so-called *euripus*), which runs at the level of the hypogea and is directed toward Gallery D, below the *Porta Triumphalis* gate, before flowing into the external sewer conduits [79]. Moreover, hydraulic bulkhead slots, allowing water retention, are documented to exist in correspondence with Galleries A and D [72]. These observations suggest that a fast and efficient filling with water could be achieved simply by limiting outflows toward the exterior, while allowing inflows through these tunnels. In this framework, it can be hypothesized that the deepest conduit served to drain the retained water. The drainage channels were obstructed starting from the 5th century [66], coinciding with the decline of the Colosseum and the Imperial Age. The drainage obstruction was probably responsible for many floods of the Amphitheatre. Attempts made by archaeologists and engineers to reactivate the drainage system were successful only when, during the 9th century, hydraulic engineers working at the Colosseum managed to reactivate a channel linked to the ancient sewer system. This confirms that large drains are fundamental to keeping this area dry.

## 4. Methods

To meet the objective of this study, log data from previous surveys were acquired from private companies, especially working at the new subway line or already collected in literature [36]. All the boreholes were drilled by the continuous coring technique, which allows "undisturbed" sampling and is appropriate to obtain complete stratigraphic logs; they often reach the depth of substratum (MVA Fm, Table 1). All data are stored in a database (WEBGIS UrbiSIT database, https://webgis.urbisit.it/webbdgt/, with data accessible under request). Where the log description was detailed enough to distinguish the lithotypes, units were associated to the stratigraphic reference frame (Tables 1 and 2). Where the description was misleading, the stratigraphy was associated to the reference frame by checking the pictures of stratigraphic logs, when available. Boreholes without a detailed log nor pictures were deleted. Apart 3 drillings, data are located outside the area of the hypogea, thus data from 2017 borehole campaign were integrated to fill existing knowledge gaps (Figure 4) about—Upper Pleistocene-Holocene Labicano Valley geometries (i.e., extension, depth and slope of the basal surface); geological bedrock depth and thickness of the encasing Pleistocene volcano-sedimentary deposits.

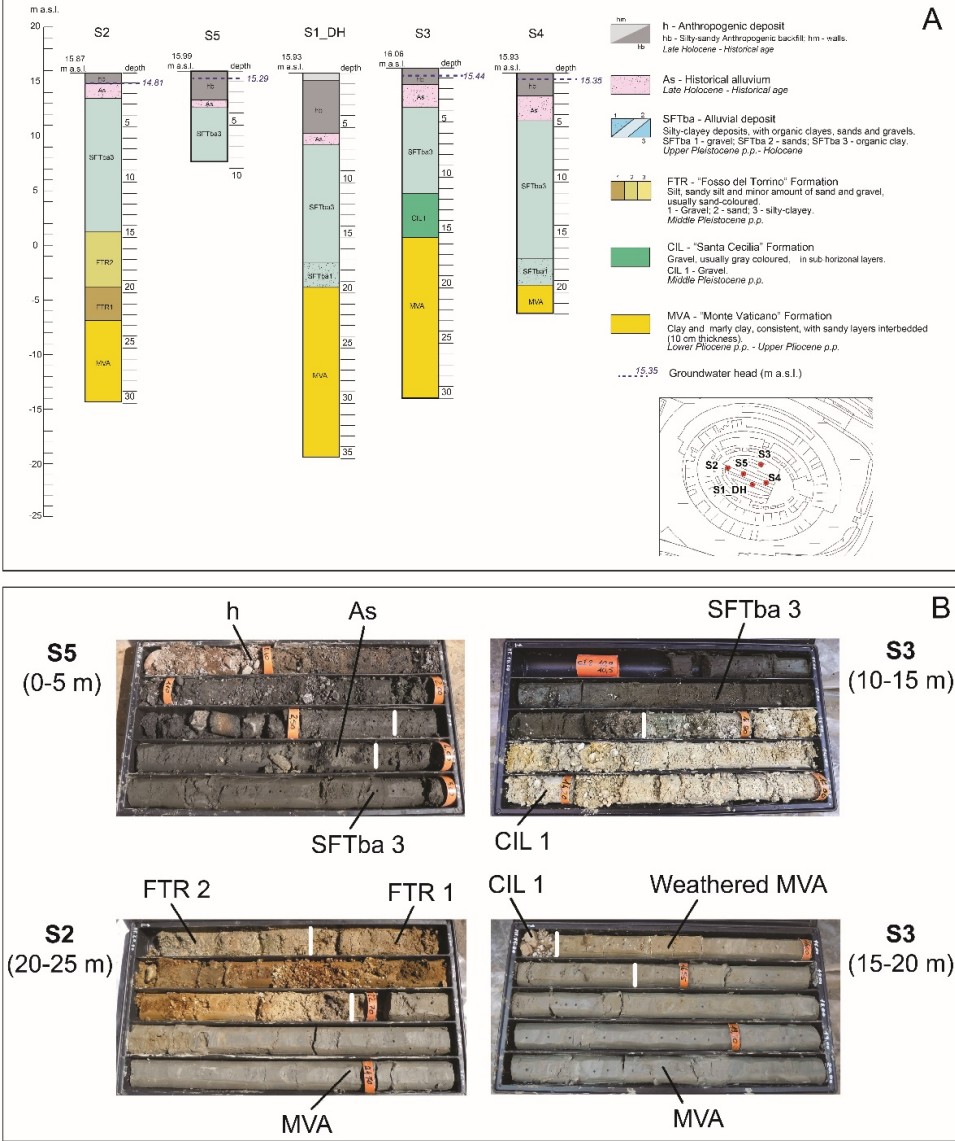

**Figure 4.** (**A**) Borehole logs (2017 drilling survey) with drilled hydrostratigraphic complexes (see also Tables 1 and 2). Well-heads are positioned at the hypogea ground surface, at approximately 16 m a.s.l. (**B**) the pictures show a sample of the cores with the drilled complexes.

*Building the 3D Model*

New and previous borehole logs (for a total of 70 logs), surface geology and interpreted cross-sections were used to build a 3D model using GeoModeller (Intrepid Geophysics©). This software uses a methodology developed in French Geological Survey (BRGM), by which 3D models are constructed using the implicit scalar potential-field method [80,81].

The method is based on the following ideas—a scalar field is defined in the space; whose gradient is orthogonal to the orientation data. The chronological succession of the formations and their relationships ("geological pile" in the software terminology) was defined, which drives the model updates [82]. Data from geological contact location points on interfaces are interpolated at the same time by universal cokriging, while the geological pile automatically drives the relationships between geological formations. The covariance of the potential field is identified from the orientation data, which can be considered as derivative of the potential. By that way, cokriging standard deviations are associated to the potential-field estimates and translated to uncertainties in the 3D model. The modelled interfaces are represented as isovalues of the interpolated field. The 3D model, centered around the Flavian Amphitheatre, covers an area of 0.211 km$^2$ (518 m × 408.5 m, Figure 5A) and extends to a depth of 100 m. A high resolution (2 m × 2 m) Digital Elevation Model, used as the top surface of the model, was built starting from the regional open access elevation point dataset ([83]). The geological boundaries of the 3D model were constrained by the lithological contacts and their orientation data, interpreted both in section and plan view. The orientation values where chosen in order to reproduce contacts typical of an area of paleovalleys and channelized sediments, where orientation data are often linked to erosional-depositional mechanism, which drive the geometries and the geological contacts of units. After each run, the modelled geology was compared to the input data. Initially, input data were not completely matched by the interpolated contact surfaces. This can be due to the fact that—1) the model needed more constraints; in this case new cross sections or a higher number of orientation data were added to the input; 2) the geology interpreted in cross-sections and, consequently, the orientation data, are affected by errors (e.g., thickness and dip of geological bodies); in this case, the sections were corrected. These operations allowed model refinements to be made.

The correctness of a 3D model is measured in terms of misfits, in other words, the percentage difference between the observed and computed depth of lithological contacts in the well logs and drawn cross-sections. A model is considered adequate when the total misfits between predictions and field data is below a certain tolerance threshold. After the first model run, results indicated 6 boreholes with misfit exceeding the 50% of their depth, out of a total of 70 imported logs. As data can be affected by uncertainty, for example, in positioning, thickness, stratigraphic interpretation, borehole logs affected by errors considered unsolvable were removed. To begin with a simple model, upper level stratigraphic sub-divisions were not considered during the first phase and only formations (i.e., higher rank stratigraphic units) were modelled. In the hydrostratigraphic model, the lowermost surface corresponds with the top of the basal aquitard, that is, the top of the MVA Fm. Middle Pleistocene CIL and FTR Fms were considered as separate complexes in the model. Indeed, although lithological differences can be scarcely appreciable, their extension, thickness and recharge areas are very different and, consequently, also their flowpaths and the groundwater contribution to the Colosseum in terms of discharge, response to rainfall and river stage fluctuation; for instance, as declared in Section 2, CIL has a tabular geometry with a wide recharge area, while FTR constitutes a paleovalley infill. Specifically, the geometric description of the gravelly-dominating layer at the base of both the Middle Pleistocene CIL and FTR Fms and its connection with the gravelly-sandy bed at the bottom of the Upper Pleistocene-Holocene Labicano alluvium, were investigated. Indeed, the extension and thickness of the contact between layers which are supposed to be highly transmissive due to their granulometric composition, can be a driving element of groundwater dynamics in the area of the Colosseum, helping understanding how groundwater is transferred throughout the modelled system. Special effort went into modelling the Labicano Valley slope, to ensure that the elevation of the valley bottom at the eastern and western margins of the model was compatible with the erosional dynamics of a fluvial system and

with the position of boreholes and cross-sections. Five new cross sections were cut, transversal to the valley axis and located in the eastern portion and at the southern and eastern model boundaries, where scarce borehole data were available. The model was then refined by adjusting the geometry of the Middle Pleistocene volcano-sedimentary formations (i.e., PTI_VGU, CIL and FTR Fms). This operation was accomplished by iteratively changing the dip direction of geological contacts in the top surface of the model. However, this process did not allow to reduce the misfit between simulated geology and observations and the surface geological contacts between PTI_VGU and CIL Fms were removed from the model input. This result reflects the large uncertainty concerning the surface geological contact, due to the presence of an anthropogenic backfill layer with a highly variable thickness that conceals any outcrops. Indeed, the maps used as model input are drawn at a basin scale on the basis of borehole data and not outcrops. The model concerns a local-scale area and comprises new boreholes, so it not surprising that the modelled geology does not match with the old maps. The release of surface topography constraints helped reduce the number of misfits.

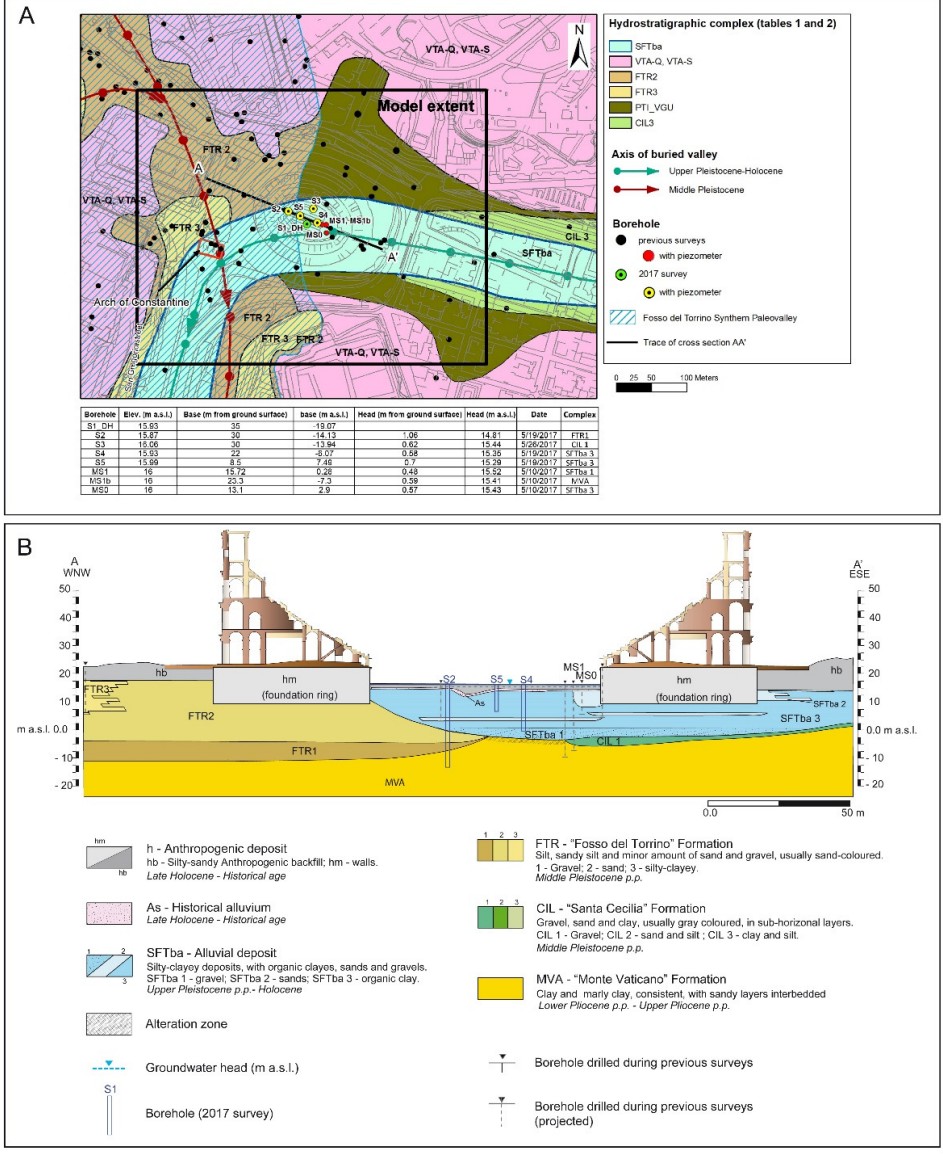

**Figure 5.** (**A**) Map of hydrostratigraphic complexes beneath the anthropogenic backfill of the study area. The table shows the head measured by piezometers in the hypogea during the 2017 survey. The extent of the 3D model is shown in the black square. (**B**) Cross-section from the 3D model. Trace in Figure 5A.

Work continued to refine the model by iteratively reviewing input data, checking the misfits after each model run and simultaneously cross-checking the sections. Results were considered satisfactory when the misfit between model interpolation and borehole data produced a maximum misfit of 5 m in 1 percent of boreholes. Then, multiple outputs were generated, such as the contouring of thicknesses and top-surface elevations. This output was exported into both vector and grid formats easy to import into other software (such as GIS and groundwater modelling software). The model was completed by subdividing the CIL and FTR Fm into CIL 1, FTR 1 and SFTba 1 complexes (i.e., lower rank stratigraphic units) in order to highlight the geometric relationships and extension of contact surfaces with surrounding complexes. The extension of formations, their boundaries and volumes were then computed.

## 5. Results

### 5.1. Stratigraphic Updates

The updates with respect to previous surveys can be summarized as follows (Figures 4 and 5):

- Anthropogenic deposits (h in Figure 4) were found in all the new boreholes, with thickness ranging between 1 and 6 m. They are composed of coarse to medium-sized fragments of bricks and stones (4–12 cm in diameter) within abundant sandy-silty matrix.
- Alluvia of historical age (As in Figure 4A,B) were also found, blanketed between the anthropogenic deposits and the underlaying SFTba unit. They are composed of sandy silt (Figure 4B), with fragments of bricks and ceramics, charcoals and terrestrial gastropods. In the model this unit is grouped with the SFTba unit.
- Fine scale texture and granulometric variations were detected inside the Upper Pleistocene-Holocene alluvium of the Labicano stream (SFTba unit). In particular, the logs provided geometric constraints (i.e., top and bottom surfaces, lateral extension) for the gravel bed at the base of the alluvium (SFTba 1 Complex), considering analogous examples of alluvial valley geometry (see References [84,85]). Given the thickness and the depth of the base of both the As and SFTba units found in logs S1_DH, S4 and S3 (Figure 4), the axis and the NW boundary of the Labicano Valley was moved southward, roughly along the S4-S1-DH line and the geological map subsequently changed. Therefore, it can be hypothesized that the WNW portion of the Colosseum foundation ring is anchored mainly in the Middle Pleistocene sands of the FTR 2 Complex, instead of being anchored to the Upper Pleistocene-Holocene alluvium of the Labicano stream, as considered in previous studies [10,11].
- The S2 borehole log (Figure 4) show that at the NW boundary of the hypogea plain, the geological bedrock (i.e., MVA Fm) reaches a depth of −8 m a.s.l., therefore approximately 5 m deeper than previously considered.
- At the uppermost portion of the MVA and CIL Fms, a 1–2 m thick alteration zone was detected, which can be ascribed to pedogenetic processes, that is, weathering because of the presence of diffuse beige-grey mottling (Figure 4B) and calcium carbonate glaebules. The weathering acted on the top of these formations prior to their subsequent burial below younger ones (i.e., FTR and SFTba units).
- The S3 borehole log (Figure 4) allowed for an interpretation of the sedimentary Middle Pleistocene sequence at the northern boundary of the hypogea ellipse as being constituted entirely of a silty-gravelly lithotype (CIL 1), instead of also comprising a silty-clayey (CIL 3) lithotype as previously interpreted. As a consequence, in the eastern sector, the bottom of the Labicano Valley is carved into a potentially transmissive aquifer, with CIL1 K ranging between 0.02–2 m$^2$/d (Table 2) and having a thickness of 4 m (Figure 4).

### 5.2. Hydrostratigraphic Updates and New Insights about Groundwater Circulation

Figure 6 shows the 3D model of hydrostratigraphic complexes beneath the anthropic backfill. In order to simplify the sketch, silty and silty sandy complexes (CIL3, CIL2 and FTR3, FTR2) were grouped together. Figures 7 and 8 show the top surface and isopach contours of anthropogenic backfill deposits and the other geological complexes, respectively. The SFTba complex is well encased in the Middle Pleistocene CIL and FTR complexes (respectively in green and brown, Figure 6b,c). The top of the basal aquiclude, that is, MVA Fm, is clearly eroded in a NS direction (Figure 8). CIL and FTR Fms directly overlay the geological bedrock; CIL Fm spreads across the eastern portion of the model. The FTR complex defines a wide NS-trending paleochannel, confined to the east by the flat-laying strata of the CIL complex. In places, apparent culminations of the Pliocene MVA bedrock ("erosive windows") interrupt the lateral continuity of the CIL and FTR complexes. As a result, the Amphitheatre is situated at a point where the Middle Pleistocene aquifers are missing or eroded and the Upper Pleistocene-Holocene Labicano Valley directly overlays MVA Fm (Figures 5B and 6).

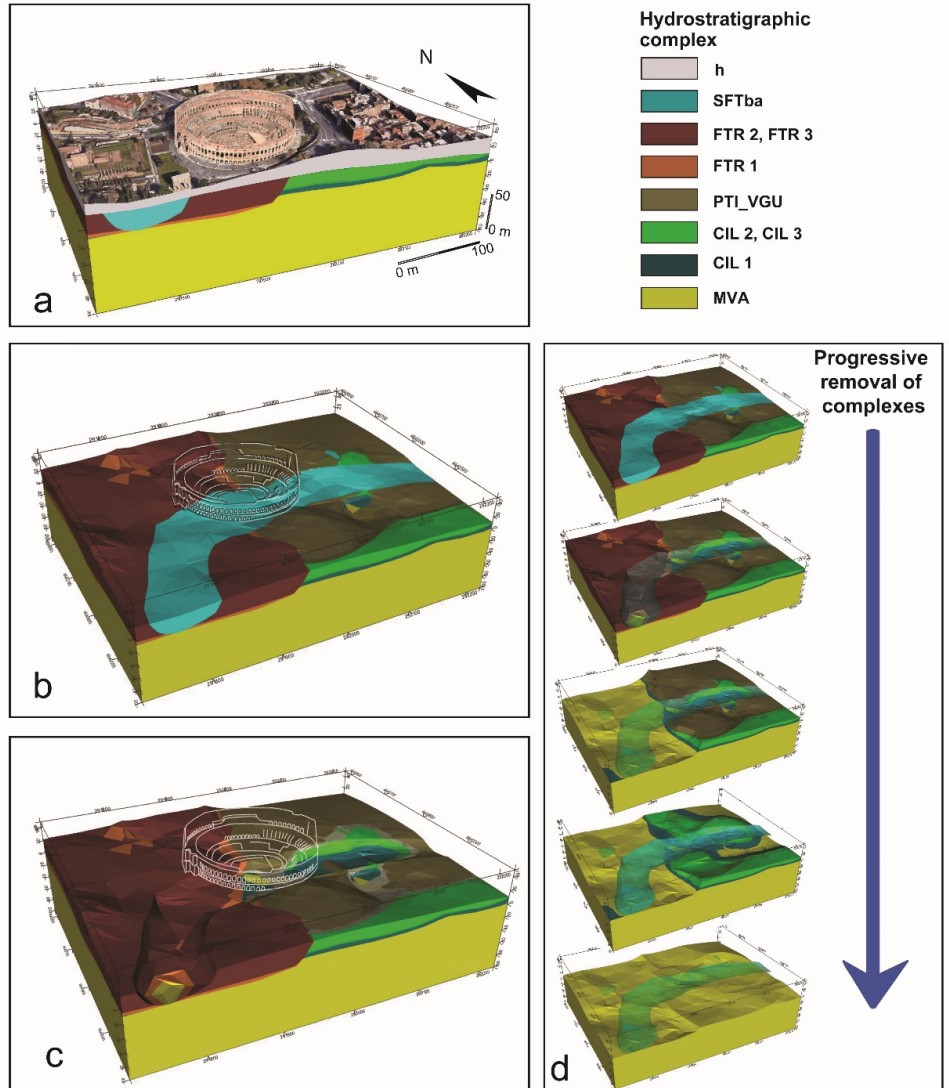

**Figure 6.** 3D model of hydrostratigraphic complexes beneath the anthropic backfill of the study area. (**a**) complete 3D view showing the anthropogenic backfill deposits (in light gray), the present topography, the Colosseum and nearby monuments and buildings (modified after Google Earth Inc.). (**b**) the same view as (**a**) without the anthropogenic deposits. (**c**) the same view without the SFTba complex. (**d**) Sequence of images showing the model with progressive removal of hydrogeological complexes.

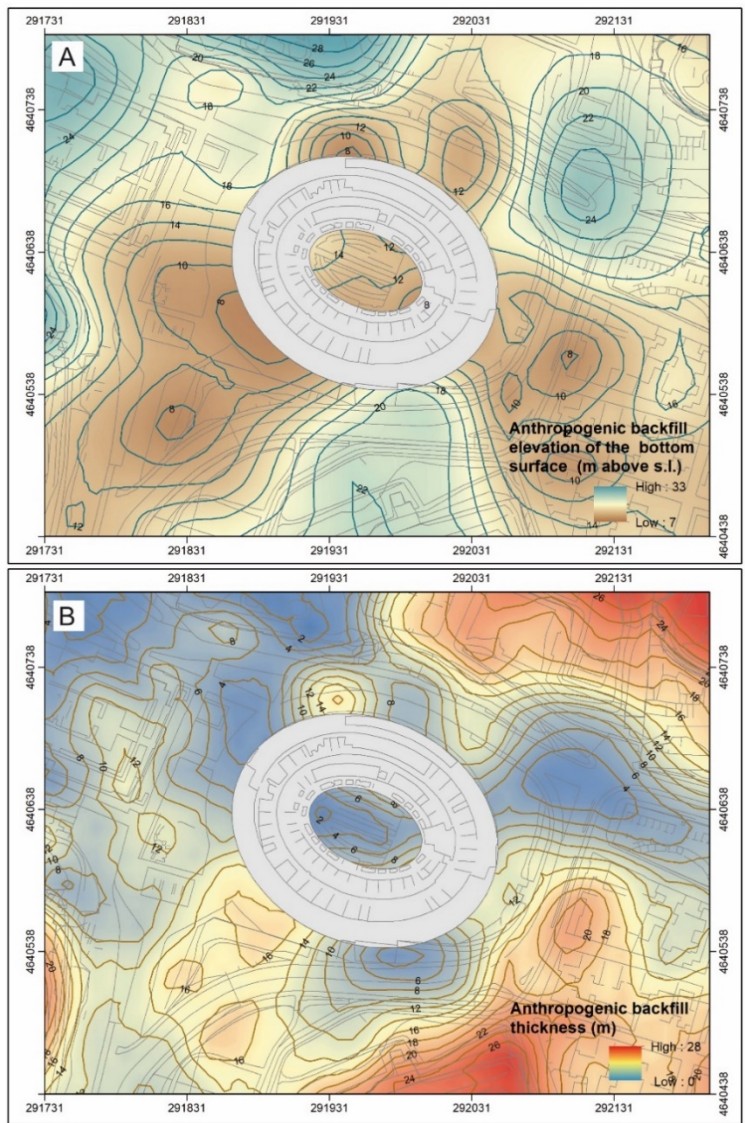

**Figure 7.** Anthropogenic backfill deposit: (**A**) elevation of the bottom surface; (**B**) thickness. The Colosseum foundations (gray ring, described in Section 3) are superimposed to the maps.

The volumes of the hydrostratigraphic complexes were calculated by exporting the centroid of voxels (with a dimension of $10 \times 10$ m). It appears how the anthropogenic backfill occupies the highest volume, in the specific area modelled. Beneath the backfill, FTR Fm accounts for the highest volume of all the geological complexes modelled (Figure 9). The contribution of VTA and CIL 2 to the model is close to negligible, due to their low volumes.

Figure 10 reproduces the model beneath the anthropic backfill. It highlights how the Colosseum (in the middle of the model area) is exactly centered at the margin of the left flank of the FTR paleochannel. Moving westward, the bedrock dips with a gentle slope toward the center of the FTR valley. Thus, in the western portion of the Colosseum, the bottom of the Labicano Valley directly overlays FTR Fm; in the eastern sector, it overlays the CIL and the PTI_VGU complexes.

Complex FTR 2 (consisting mainly of sands with interbedded silty-clayey portions of FTR 3) reaches a thickness up to 40 m, while the thickness of the silty-gravelly basal layer (FTR 1) ranges between 3–10 m, decreasing from the center of the site westward. To the east, the CIL Fm shows prevailing silty-gravelly (CIL1) textural composition, instead of silty facies (CIL3) as it was believed before. This suggests that notable groundwater inflows toward the Colosseum can arrive from the

Oppian and Caelian hills (N and S flanks) and from the Labicano stream valley (East flank) due to the high transmissivity of the CIL Fm. Moreover, the Holocene valley appears to be excavated into highly permeable sediments, except at the center of the Amphitheatre, where it directly overlaps the substratum. The Sftba 1 complex, reaching a maximum thickness of 3 m at the center of the valley, is thus hydraulically connected with the gravelly complexes CIL 1 and FTR 1.

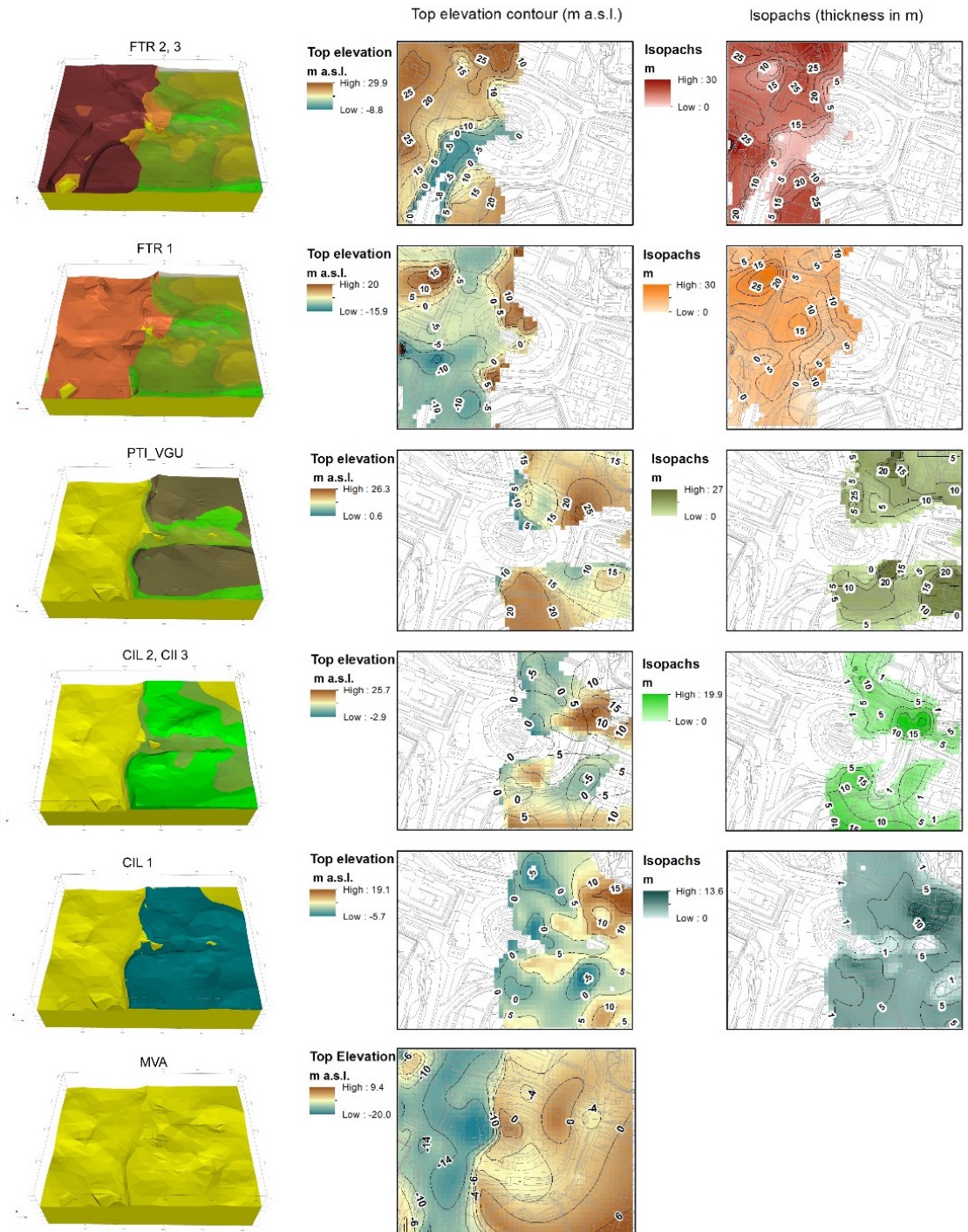

**Figure 8.** Top surface elevation and isopach contours of modelled formations.

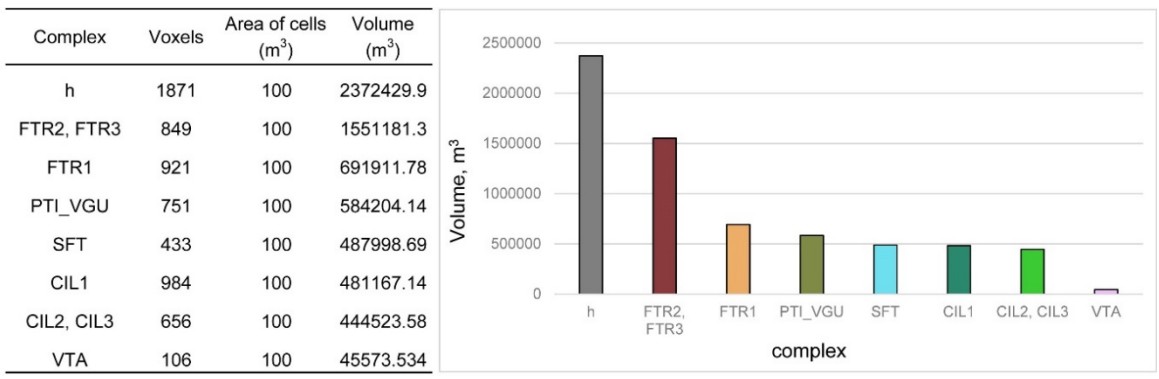

| Complex | Voxels | Area of cells (m³) | Volume (m³) |
|---------|--------|--------------------|-------------|
| h | 1871 | 100 | 2372429.9 |
| FTR2, FTR3 | 849 | 100 | 1551181.3 |
| FTR1 | 921 | 100 | 691911.78 |
| PTI_VGU | 751 | 100 | 584204.14 |
| SFT | 433 | 100 | 487998.69 |
| CIL1 | 984 | 100 | 481167.14 |
| CIL2, CIL3 | 656 | 100 | 444523.58 |
| VTA | 106 | 100 | 45573.534 |

**Figure 9.** Calculation of volumes for complexes in the study area.

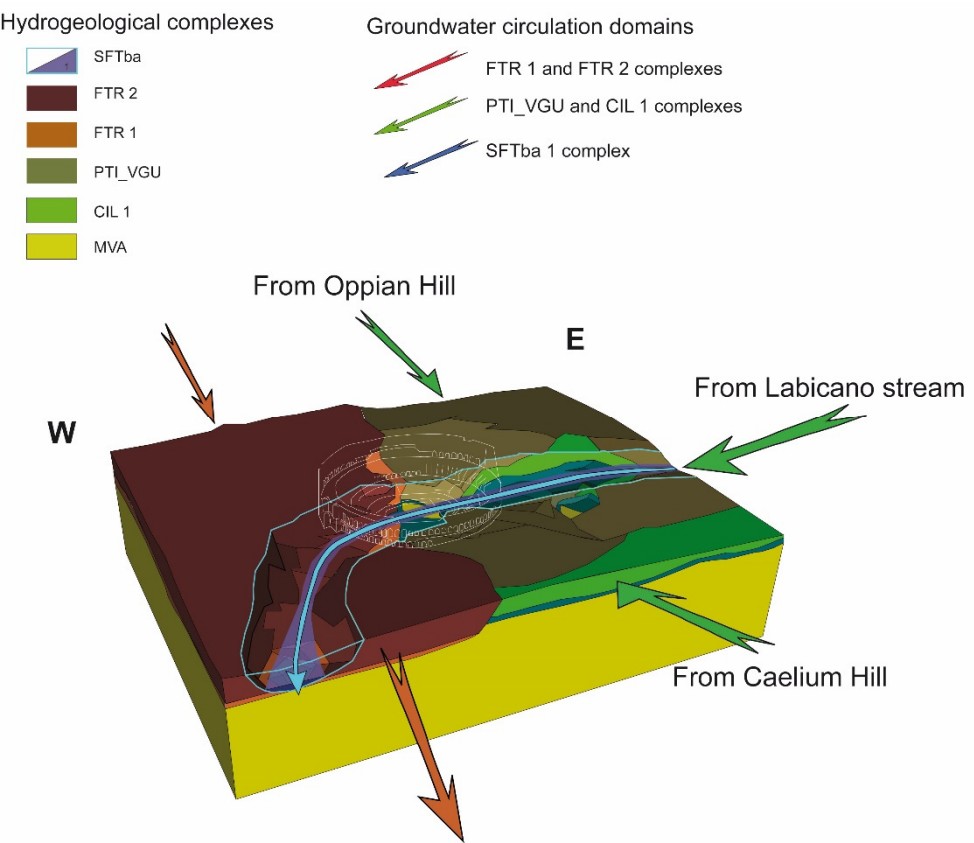

**Figure 10.** 3D model beneath the anthropic backfill and deep groundwater circulation: arrows represent the groundwater flow direction through FTR Fm (orange arrows) and through CIL and PTI_VGU Fm (green arrows). See hydrostratigraphic frame in Table 2 for abbreviations.

Piezometric measurements between May and June 2017 in new and old boreholes (see table in Figure 5A), showed that aquifers in the CIL 1, SFTba 1 and SFTba 2 complexes present very similar heads, ranging from 15.29–15.52 m a.s.l. Differently from the easternmost boreholes, piezometer S2 shows a net lower level (14.81 m a.s.l.) with respect to the others; S2 presents a different stratigraphic log, with the FTR 1 complex below the alluvium. This confirms that the groundwater head decreases toward the NW portion of the hypogea, with flow directed from E-W, from higher to lower potential. Differences in hydrodynamics were also observed to affect S2. Indeed, during the storm event of 19 May 2017 (47.4 mm recorded at the Collegio Romano pluviometer, see Figure 2A), head levels in piezometers S4 and S5 showed a 40 cm increase in 24 h. Curiously, a different situation was recorded at piezometer S2—a head rise of 40 cm (from 14.7–15.1) was recorded only 13 days later. The delayed

head rising could be related to different recharge areas and connections with the Tiber River or the minor diffusivity of the aquifer, with respect to the other monitoring points. What is clear is that the hydrodynamics of the rainfall-runoff-groundwater infiltration process acts differently in the FTR formation than in the CIL and Holocene Fm.

## 6. Discussion

The 3D model made it possible to define hydrostratigraphic constraints for the groundwater conceptual model (Figure 10) and to formulate hypotheses about flow dynamics during storm events. Complexes CIL 1 and FTR 1 are both confined aquifers in the study area, with a head approximately 40 cm from the ground surface (close to the head of alluvium). This suggests the capability to transfer groundwater in an upward direction, toward the uppermost aquifers (Figure 11).

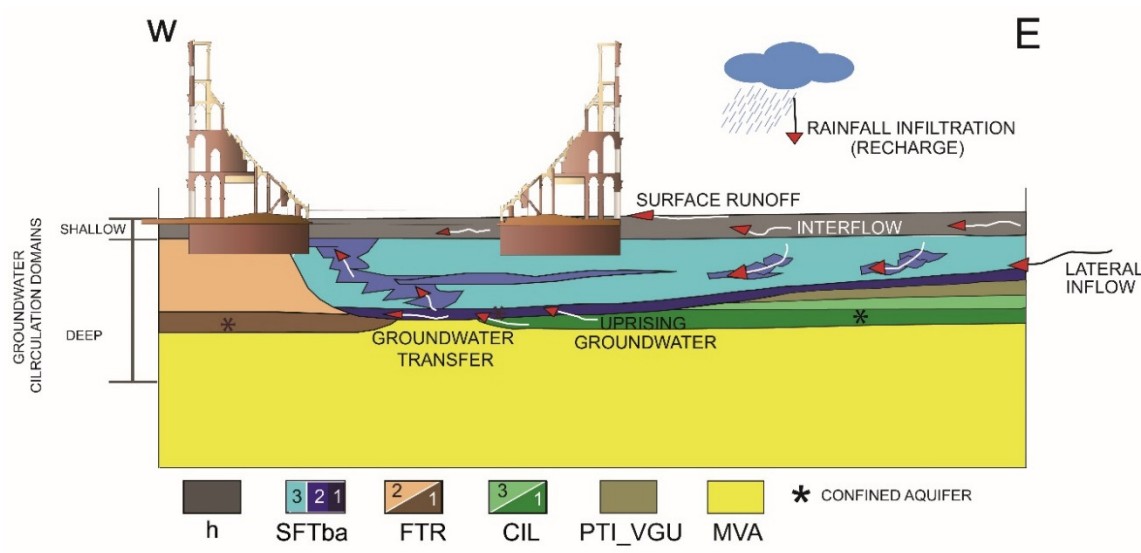

**Figure 11.** Groundwater Conceptual Model. Not to scale.

The gravelly layer corresponding to complex Sftba 1 can thus be fed by groundwater rising up from the CIL 1 and FTR 1 complexes, depending upon the piezometric head field, mainly through the sandy lens interbedded in the alluvium (SFTba 2). As a consequence, two domains of groundwater circulation can be distinguished. A shallow unconfined circulation (I) is hosted in the backfill deposit and in the uppermost portion of the SFTba complex. This aquifer is fed primarily by both rainfall infiltration and overland flow along the hydrological basin of the Labicano stream; it flows toward the Colosseum before draining southward into the Labicano valley itself. A deep circulation (II), hosted primarily in the CIL1 aquifer, directed from east toward the center of the Colosseum, where it is transferred to the FTR paleochannel. The bottom of FTR Fm, lower than that of CIL Fm and the FTR head, lower than that of CIL, facilitate groundwater transfer toward FTR. At the center of the Amphitheatre, where the lateral continuity between complexes CIL1 and FTR1 is interrupted, the narrow SFTBa1 complex deposit provides the hydraulic connection of the deep circulation from upgradient (E) to downgradient (W). Thus, the deep groundwater flow direction suddenly turns from EW to NS, following the paleochannel axis. On the one hand this setting suggests the existence of a constant groundwater inflow toward the area of the Colosseum, guaranteed by the highly transmissive CIL1 Fm aquifer, with flow locally directed from E-W; moreover, exterior outflow is ensured by the drainage operated by the FTR1 complex.

In this manner, both surface water and groundwater can quickly flow into the Amphitheatre and be rapidly drained. Anthropic modifications to the hydrological system exploited this particular setting, tending to manage and control inflow-outflow rates. The annular surficial channel in the hypogea (*euripus*) maintains shallow circulation at ground level, ensuring drainage toward the exterior.

Under a normal hydrological regime, the large underground conduits are able to drain both shallow waters (i.e., surface runoff, leakage from aqueducts) and groundwater rising up from deep confined aquifers. In the case of intense storms, the flooding of the Tiber River or a combination of the two, the equilibrium breaks down; the increased runoff and groundwater recharge generated by rainfall implies an increase in the groundwater head in both shallow and deep aquifers, with a consequent increase in rising groundwater from deep aquifers. Moreover, when the Tiber floods, sewers have a reduced ability to drain groundwater and are quickly overwhelmed. Following this conceptual scheme, floods observed during storms could be ascribed to increased infiltration and runoff rates, combined with a high river stage. This is in agreement with the flooding of the hypogea during the 21 October 2011 storm event, when a Tiber river flood was observed, reaching 7.76 m at the Ripetta Gauge (Figure 2A).

Despite these conceptual updates, the understanding of groundwater inflows from the small hills surrounding the Colosseum through the eastern, southern and northern galleries must be still clarified. Also, it is not possible to calculate the timing of the rainfall-runoff process from each single inflow direction. The role of anthropogenic backfill need further investigation. Indeed, it represents the major volumetric contribution to the model and is the uppermost complex, thus, it directly receives stormwater and connects the two domains of surficial water and groundwater; thus, its role in modulating the effects of storms should be clarified. It is fundamental to obtain a complete scheme of the contribution made by all sources of water, quantifying the maximum drainage capacity of artificial drains. Long-term piezometric monitoring, over at least three hydrological years, recording hourly levels and chemical-physical variations, may provide elements able to solve issues related to groundwater dynamics or relationships between rainfall and piezometric levels or help clarify the condition of aquifer confinement. A suggestion for a monitoring network is shown in Figure 12.

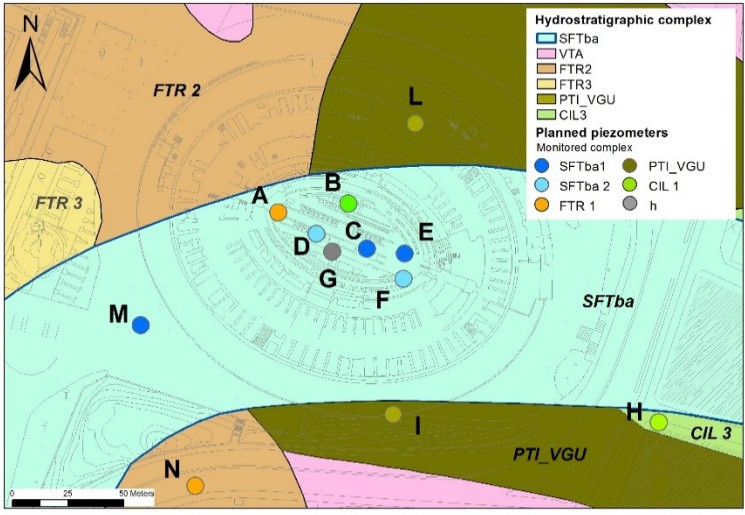

**Figure 12.** Suggested piezometric monitoring network configuration.

Inside the Amphitheatre, it comprises 2 piezometers within the Middle Pleistocene aquifers (FTR 1 and CIL 1 complexes, A and B, respectively), 4 piezometers in the Holocene alluvium, both in the basal gravel (SFTba 1, C and E) and sandy layers (SFTba 2, D and F) and one piezometer in the anthropogenic deposit (h, G). Other monitoring points should be located outside the Colosseum, to clarify the mechanism of upgradient inflow and downgradient outflow and the contributions of all aquifers in complexes—CIL 1 (H), VGU_PTI (I, L), SFTba 1 (M) and FTR 1 (N). This piezometer configuration ensures the monitoring of both deep, productive aquifers and shallow aquifers.

Monitoring data can be then used to update the conceptual groundwater model and to develop a numerical model, in order to confirm or discard hypothesis about the inflows contributions and clarify the timing of aquifer response during floods and storm events. Calculating the water contributions is the first step for planning mitigation measures, both structural (such as the reactivation of historical

drains or the activation of new drains) and non-structural (such as the precautionary closing of the arena to tourists). Calibrated under actual hydrogeological and hydrological conditions (i.e., actual rainfall regime, inflows, outflows, piezometric levels), the numerical model could then be run using different input data and boundaries, such as different river stages or higher water table elevations (this latter would most likely approximate the condition during Antiquity), to support hypotheses about past hydrological conditions.

## 7. Conclusions

The aim of this work was to evaluate aquifer connectivity using a 3D hydrostratigraphic model, furnishing a conceptual scheme of groundwater inflows toward the Colosseum. The geological model helps optimize the management of stratigraphic information; 3D visualization speeds up the process of stratigraphic setting evaluation, allowing for verifications of existing geological maps and sections. The model was helpful for detecting geological contacts, calculating volumes and thicknesses and evaluating geometric relationships between hydrostratigraphic complexes. The 3D model, together with historical-archaeological information and observations, allowed for the development of a conceptual model describing the dynamics of groundwater and surface water inflows toward the Amphitheatre. Two groundwater circulation domains were distinguished—a topmost, shallow domain, hosted into the uppermost portion of the SFTba Fm and the anthropogenic backfill deposit, comprises mainly of local rainfall infiltration and overland flow; and a deeper domain, constituted by groundwater flowing into the Middle Pleistocene aquifers (CIL and FTR Fm). The highly transmissive alluvial gravel basal bed (Sftba 1) acts as a preferential drain for groundwater rising up from the confined Middle Pleistocene aquifers, ensuring the hydraulic continuity between the CIL Fm and the NS-trending FTR Fm, even where their lateral contact is interrupted. An upward groundwater transfer between the deep and shallow domains, cannot be ruled out and likely occurs through the sandy portions of the alluvium (SFTba 2). Thanks to this double system, both surface water and groundwater are collected toward the center of the Colosseum, before being drained to the west and then southward. During intense storms, increased runoff and piezometric head can result in an increased upward flow. When the Tiber floods, the channels and sewers can be overwhelmed and the increased inflow cannot be drained, resulting in flooding of the hypogea. At the present stage, the 3D model furnishes a robust base useful for implementing a detailed local-scale groundwater conceptual model. However, the complex water dynamics, together with the system's response to intense storms, cannot be fully understood without a proper head levels monitoring campaign, coupled with rainfall and river stage monitoring. The 3D model was useful for detecting the main gaps in knowledge about the groundwater system and optimizing the placement of monitoring piezometers. Once existing data gaps have been resolved following proper monitoring, the conceptual model can be refined. Further studies can then involve the development of groundwater numerical models, in order to verify the hypothesis of inflow-outflow dynamics, speeding up the process of water management optimization. This is particularly relevant for:

- ensuring drainage under normal flow conditions necessary to the correct preservation of the Colosseum.
- planning flood risk mitigation measures, both structural (such as the activation or reactivation of artificial drains) and non-structural (such as the precautionary closing of the area to tourists).

**Author Contributions:** Conceptualization, C.D.S., M.M. (Marco Mancini); methodology, C.D.S.; software, C.D.S., G.P.C.; validation, C.D.S., M.M. (Marco Mancini); formal analysis, C.D.S., M.M. (Marco Mancini), G.P.C.; investigation, C.D.S., M.M. (Marco Mancini), M.S., A.R.; resources, G.P.C., A.R.; data curation, C.D.S., F.S.; M.M. (Marco Mancini); M.M. (Massimiliano Moscatelli); writing—original draft preparation, C.D.S.; writing—review and editing, C.D.S., M.M. (Marco Mancini) and M.M. (Massimiliano Moscatelli); visualization, C.D.S., F.S., M.S.; supervision, R.R. All authors have read and agreed to the published version of the manuscript.

**Funding:** This research received no external funding.

**Acknowledgments:** Authors would like to sincerely thank the reviewers for all the invaluable suggestions which allowed us to significantly improve the content of the manuscript.

**Conflicts of Interest:** The authors declare no conflict of interest.

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
