# Peer review of "A 3D Geological Model as a Base for the Development of a Conceptual Groundwater Scheme in the Area of the Colosseum (Rome, Italy)"

_geosciences, doi:10.3390/geosciences10070266_

Round 1

Reviewer 2 Report

I appreciate that the authors have made a considerable effort to improve the paper by adding more explaining text in order to avoid misunderstandings. I acknowledge that the text in many places has been improved.  

However, I still have reservations on these subjects:
• The paper is still too long and, in my opinion, it contains many descriptions that are not relevant for the scope of the paper.
• A mixing of geology and hydrostratigraphy throughout the text is still present, exemplified by the figures 2, where the units are ‘geologic’ and Figure 4, where the same units are ‘hydrostratigraphic’.
• The descriptions of the modelling procedures have been improved, but also extended. In my opinion, many of the authors’ descriptions are about ‘workarounds’ dictated by the modelling software. It is necessary for the reader to know the basic procedures of the modelling process, but as the model does not contain very complex geology, maybe lengthy descriptions of how to avoid fx ‘unrealistic interpretations’ by the software could be shortened or left out.
• I still find the section on surface hydrology and archaeology too long and not necessary for the scope of the paper. It should be significantly reduced in size.
• The Discussions section still present a lot of hypotheses and guesswork about groundwater flow; not surprising, because it is difficult to describe the groundwater flow without a numerical hydrological model.

My reservations mentioned above are not critical and I will leave it to the editor to decide on further corrections of the paper.

Kind regards

Author Response

I appreciate that the authors have made a considerable effort to improve the paper by adding more explaining text in order to avoid misunderstandings. I acknowledge that the text in many places has been improved.  

However, I still have reservations on these subjects:
• The paper is still too long and, in my opinion, it contains many descriptions that are not relevant for the scope of the paper. 

Answer: following the suggestions specified in the comments below, we deleted some sentences which we found unnecessary.

  • A mixing of geology and hydrostratigraphy throughout the text is still present, exemplified by the figures 2, where the units are ‘geologic’ and Figure 4, where the same units are ‘hydrostratigraphic’.

Answer: Dear reviewer, we just would like to remark that we used the tables 1 and 2 as a key for “translate” geology into hydrostratigraphy. In table 1, the geological formations are broken into lithotypes; in some case, the single lithotype correspond to one distinct hydrostratigraphic complex, due to its peculiar permeability properties. This is for example the case of FTR1, FTR2 and FTR3. Otherwise, in some cases lithotypes are grouped together into hydrostratigraphic complexes: for example, AEL, VSN2, VSN1a and VSN1b are grouped into the hydrostratigraphic complex VTA-Q.  While figure 2 reports a geological map and section from literature works and represent the geological formations, the subsequent figures represent the hydrostratigraphic units. In the case of Figure 4, all the represented hydrostratigraphic complexes are coincident with the lithotypes: SFTba1, SFTba 3, FTR1 and 3, CIL 1. Otherwise, other figures (for example figures 5, 6, 8, 10) report hydrostratigraphic complexes which not necessarily coincide with lithotypes:  for example, PTI-VGU, which results from the grouping of many units.

  • The descriptions of the modelling procedures have been improved, but also extended. In my opinion, many of the authors’ descriptions are about ‘workarounds’ dictated by the modelling software. It is necessary for the reader to know the basic procedures of the modelling process, but as the model does not contain very complex geology, maybe lengthy descriptions of how to avoid fx ‘unrealistic interpretations’ by the software could be shortened or left out.

ANSWER: According to this suggestion, we deleted some sentences in section 4.1, corresponding to about 7 rows, which we found unnecessary.

  • I still find the section on surface hydrology and archaeology too long and not necessary for the scope of the paper. It should be significantly reduced in size.

ANSWER: Following your suggestion, we did an additional revision of section 2, deleting some sentences redundant and unnecessary. With respect to the archaeological section, we would like to remark that our description tends to give to the reader a reviewer of the literature describing the groundwater contribution to the arena from the hills and valleys along the ellipse axes. Since there are many uncertainties and there is scarce knowledge about archaeological remains of drains and cisternae, further coherent numerical models should contain hypotheses about how groundwater can be delivered to the arena, simulating the setting of the man-made drains and channelized streams; although the geology of the area is simple, inserting man-made drains can be a crucial step to obtain results which can confirm or discard the archaeological literature works. For these reasons, we decided to maintain section 3 almost complete as in the original manuscript the section of the anthropogenic modification of the hydrological setting.

  • The Discussions section still present a lot of hypotheses and guesswork about groundwater flow; not surprising, because it is difficult to describe the groundwater flow without a numerical hydrological model. 

ANSWER: Dear reviewer, as you mention, the discussion of a conceptual model here only represents hypotheses, since no numerical simulation were made. However, in our opinion, this discussion is necessary in the phase of numerical model setting up, when the consistent hypotheses are verified or discarded by the numerical results.

My reservations mentioned above are not critical and I will leave it to the editor to decide on further corrections of the paper. 

Kind regards

Reviewer 3 Report

Thank you for the great effort you put into this first round in order to solve my comments. I think your paper has now reach the target.

Best 

Author Response

Thank you for the helpful suggestion which allowed to improve the quality of the manuscript.

This manuscript is a resubmission of an earlier submission. The following is a list of the peer review reports and author responses from that submission.

Round 1

Reviewer 1 Report

This is an interesting and well-written paper, presenting a detailed model of an important part of Rome, to support subsequent analyses for water management. The paper uses new local field information and a good review of literature relevant to the site, with some reference to wider international literature on 3D digital geological modelling, a rapidly developing field of study.

Although not yet fully published, the authors may be interested for subsequent work (if they are not already aware) of the forthcoming publication (in press, for publication in 2020): Applied Multidimensional Geological Modelling: Informing sustainable human interactions with the shallow subsurface [A. Turner and H. Kessler, Eds.]. Wiley. (Includes two of my contributions - a chapter on process modelling, and a short case study on archaeology and hidden rivers.)

I have no substantial issues with either the content, methods, or presentation of the paper, and would recommend publication of the paper subject to the minor comments below, mostly on clarification.

lines 364 ff: The analysis of correctness of a 3D model assumes a particular modelling methodology, and how the borehole observations are used to constrain the model structure. It would be helpful to provide some brief background on how the GeoModeller package does this.

line 368: a precision for misfits of 10m seems high, compared to the vertical scale of the geological structures.

lines 371-374: this sentence is unclear. A little more description of the modelling approach may help to clarify this. Typo: 'tacking' should read 'taking'

lines 415-416: following the comment above, the misfit tolerance is correctly reduced across most of the model. It may be worth a brief statement on whether the remaining erroneous sites are critical to the hydrogeological interpretation (eg whether they may represent locations of hydraulic connection)

Some really interesting 3D and conceptual cross-section images are presented, showing clearly the relation between the geological and hydrogeological interpretations, and the Colosseum structure. On Figure 10, the label "OVERLAND FLOW" appears to relate to a line that looks as though it represents interflow (shallow subsurface flow).

Reviewer 2 Report

Overview

The paper describes the development of a 3D geological model intended as a basis for a conceptual hydrogeological model (scheme) for a small local area around the Colosseum, Rome. The paper reviews earlier geologic/hydrostratigraphic modelling work in the area and describes the building of a 3D model using new borehole data. The model results are used to describe the hydrogeological/hydrological setting hypothesized from the new model and the need for further data collection and modelling.

The paper conveys an interesting insight into local hydrological hazards around one of the Worlds most famous archaeological landmarks. Obviously, there is an urgent need for 3D mapping and modelling in order to be able to keep the visitors safe and to help preserving the site. The paper represents one of the first steps, and the authors enthusiastically evaluate existing models and data, builds a new, revised 3D model and sketches a strategy for further work. The paper provides interesting details about the current hydrogeological situation at the site based upon an impressive list of references.

However, I have several reservations, because in my opinion the paper does not add enough new facts to be sufficiently interesting. I find that a major revision of the paper is needed. I think that the authors should trim the paper to an easy read that has fewer focus points. I hope that the authors in that process will focus on new important knowledge that was not present before – knowledge that now enables them to suggest specific further work.

In the following, I list a number of comments – both generally and specifically - for the authors to consider. I hope that my comments will be received in the positive manner in which they are intended, and I hope that the authors will be encouraged to do further work on the paper.

General comments

  • The 3D modelling led to new hypotheses but unfortunately not to firm hydrogeological or hydrological conclusions. To that end – and no surprise - a numerical groundwater model is needed. The authors know that, and the paper ends with a list of needed future work.
  • The outline of the paper is fine per se, but as it does not bring about any hydrological conclusions based on a numerical model, its value diminishes. A story about building a geological 3D model and a conceptual hydrogeological model for the Colosseum could be an interesting read, but 29 pages is way too much. It could easily (and should) be reduced in size to about half of the present page count. The authors’ high ambitions to include archaeology into the geo/hydrology story is one of the reasons for the lengthy text. I perfectly understand why, but the difficult thing is to cut the story ‘to the bone’. However, I recommend that the authors do exactly that.
  • A general comment is about definitions. Throughout the paper the terms ‘geological model’, ‘conceptual scheme/conceptual model’, ‘hydrostratigraphic complex’, ‘hydrogeological scheme’, ‘hydrological model’, ‘hydrogeological unit’ and so on are used, but no definitions are presented. In Figure 1 for instance, alluvial, detritical-alluvial and volcanic ‘units’ are referred to as ‘hydrogeological units’. This is geology, not hydrogeology/hydrostratigraphy. The same is the case in Table 2; columns with ‘hydrogeological units’ and ‘hydrogeological complexes’ contain geology. Not hydrogeology. There are many other examples, and there is no clear cut discrimination between what is geology (the sediments themselves, the lithologies, the structural build of the subsurface), what is hydrogeology (the aquifers, the aquicludes/aquitards, the hydraulic head) and hydrology (the groundwater/surface water flow systems). I am fully aware that there are many ways you can work with geology and hydrology, but in order to avoid misunderstandings, definitions are welcomed. In my world, a 3D geological model is the first step, afterwards the geological layers are converted into 3D hydrostratigraphic units which then act as input for the hydrologists and their numerical modelling.
  • In my opinion, the introduction includes a lot of unnecessary descriptions of general uses of geological models that take away focus from the aim of the paper.
  • The sections describing how the 3D model was made (Chapter 4) left me without an overview of what ‘the previous model’ of the area looked like and which changes were made during the revision of it; was the general picture maintained or revised, and were the changes minor? I would guess so, because only 7 boreholes located inside the Colosseum comprise the new data for the updated and revised model. I may have overlooked something, but anyway, it is not clearly described.
  • The way the model was build is described, but it is more on how iterations, simulations and ‘fittings’ are made in the software and not so much about in which way actual borehole data has been used. It appears to me, that the software used is optimal for 3D models based on measurements of outcrops, whereas it does not seem to be the best choice for 3D models based on borehole data as the present case.
  • The Results and the Discussions section presents a lot of hypotheses and speculation, and in my opinion, it clearly exposes the need for a numerical groundwater model. I recommend that the sections are shortened and the text focus on what is new compared to the previous model and which possibilities for hitherto unrecognized hydrological connections between the hydrostratigraphic units the new findings point to. These focus points can then be used to outline precise specifications for future work.

Specific comments

Line 41: Please reconsider the use of ‘terrain’ to describe sedimentary formations here and throughout the paper.

Line 42: ‘Spatial relationships between geological formations and different facies within a single formation are responsible for groundwater paths’: What about hydraulic head?

Line 45: ‘Developing a 3D geological model using a specific software brings many advantages.’ What is meant by that?

Line 46: ‘Interpreted geology can be compared to field data in a 3D environment allowing the modeler to quickly detect errors or inconsistencies.’ I am not sure of what is meant here. Field data is normally used to interpret geology. A geological model is BASED on data; NEW data can be used to revise an existing model.

Line 49: More explanation of how the model is made is needed here. ‘Iterative processes, computations and automatically updating’ sounds like a data-driven approach where the software is in control? It does not seem as the best choice for a very small model based primarily on borehole data.

Line 52-62: One of the sections that I find unnecessary.

Line 98: ‘Comprehending surface water-groundwater dynamics and interaction together with anthropic modifications of the hydraulic system, is fundamental to understanding flood mechanisms.’ True, but as far as I can see, the paper does not include any suggestions for a mapping of just that. The anthropogenic layers are mentioned as a sort of ‘black box’ that is important, but unfortunately not mapped.

Line 111: Here, and elsewhere in the text, ‘geological hazard’ is mentioned. Please define. A flooding is in my opinion not a geological hazard but a hydrological one.

Line 124: The fluvial valleys are carved out ‘and in some cases buried under older geological formations’. Is there an error in the sentence here? It seems to be against ‘geological principles’!

Line 170: Please explain what is meant by ‘a single groundwater circulation’.

Line 214: ‘Semi-outcropping’ and ‘semi-unconfined’? ….when beneath the anthropogenic layer….? This is difficult to understand and needs more explanation. Again, the anthropogenic layer seems to be a ‘black box’.

Line 223: Why is it necessary to calculate lithological volumes?

Line 236: Here a 3D model is mentioned. Which one? Made by whom?

Line 238: Table 1. Is this stratigraphy the generally accepted one? Can sediments in the area easily be related to this scheme? Are there any problems with it or is it accepted ‘as is’? Please describe in the text.

Line 254-323: All this information about surface hydrology and archaeology is very interesting, but as far as I can see, it has no relevance in this paper. It will have, if the paper was about a numerical hydrological model that encompassed both surface water and groundwater.

Line 325: How was the previous surveys analysed and which revisions were made?

Line 359: ‘As required by the software….’: Geological boundaries were constrained by dip and azimuth data…..? Where did these data come from? It seems as if old maps comprised the data here….!? ‘After running the model, the interpreted borehole logs were compared to the modelled geology’: This makes no sense to me; the boreholes ARE the data. Old maps are derivates of old (uncertain?) data. But maybe I misunderstand something here….

Line 374: ‘Due to scarce model constraints, misfits at model boundaries were considered negligible.’: Please explain; hard to understand.

Line 381: ‘Middle Pleistocene CIL and FTR Fms were considered as separate complexes in the model, although lithological differences can be scarcely appreciable’: Is that because the ‘stratigraphic scheme’ dictates it? The following lines do not clarify….

Line 395: ‘valley thickness’: A valley has no thickness, but the infill has.

Line 407: What is an ‘unsatisfactory result’? Please explain.

Line 408: ‘This result reflects the large uncertainty concerning the surface geological contact, due to the presence of an anthropogenic backfill layer…’: Yes, it is very important to map the anthropogenic layer. Is there a reason why this has not been done?

Line 410: ‘The release of surface topography constraints…..’: What does that mean?

Line 412: Please explain ‘simulation’ more thoroughly.

Line 428-433: Is this concluded from one borehole only?

Line 446-452: Difficult to understand; should this be moved to section 4?

Line 474: On Figure 4 instead of Figure 6?

Line 477: Reduce Figure caption and include in text instead.

Line 492: Why calculate volumes? And why use that extreme precision of the values?

Line 503: Because of a small volume, the layers are not important hydrogeologically? How is that? In ‘my world’, layers with minor thickness can be very important. Only numerical groundwater modelling will show if that is the case or not.

Reviewer 3 Report

The authors develop in the paper,  a geological model of great interest for two different, but in any case correlative reasons, on the one hand the presence of a monument of world archaeological importance and on the other hand for the complex geological history and  consequent geomorphological evolution of the "containment site" of the Colosseum; history which in some way contributed with anthropic actions to modulate the Roman settlement.

The perspective to setup a groundwater conceptual model in a scenario of quite high flood hazard, drive all the efforts of authors.

Following are some questions-considerations and simple notes for the authors:

33: "..of sediment and rock.." --> "...of sediments and rocks.."

45-46: "Developing a 3D geological model using a specific software brings many advantages." I believe in 3D geomodel apart from the role of some specific software (the black box) so I suggest to remove the phrase because, in this regard the importance of 3D model is clearly underline in the former paragraphs, or better to rewrite it in order to connect/introduce the following paragraph.

46: "Interpreted geology can be compared to field data.." --> "Interpreted geology can be compared to field and subsurface data from direct (eg. field) and geophysical approaches."

64: "..as a base for alternative..." --> what means alternative?

67: "In this regard, [7] highlighted how 3D models of archaeological layers provide much more information than standard approaches based on 1D or 2D mapping, permitting an understanding of archaeological layering by means of volume calculation and the simulation of archaeological formation and deformation processes..." --> I don't believe that the work of Barcelò and colleagues [7] may be used as  valid reference here, as they provided how a 3D model of archeological fieldwork may useful for mapping very shallow environments, not your deeper model. I suggest to remove it.

249: in caption of fig.2 "Geological complexes..." what's mean complexes?

254: in the title of cap.3 you use both the term "Hydraulic" as well as "Hydrological", Why? And following it why not use instead the "Hydrological Setting and its Anthropogenic Modifications"?

290-291: "However, the discharge of the Labicano, corresponding to an average of 1 l/s [74], is clearly insufficient to justify the dimensions of the external sewer conduit." This measure is the actual discharge, correct? How much it adequately represents the historical discharge and so the dimension of sewer?

325: "..log data from previous surveys were acquired and analyzed.." Given the importance of these data please supply if possible, more info about these data apart their location. Seems to me of extremally interest know something about them because, their role in the modelling phase.

342: "Despite its short duration...to the definition of the groundwater lateral and

vertical gradient.." --> what method was do estimate lat and vert gradient used?

367-369: "At first, a maximum allowable distance (precision) of 10 m between the contacts observed in boreholes and those in the model was used; results indicated 6 boreholes with misfits exceeding the tolerance, out of a total of 70 imported logs" --> Why, in which manner you choose the value of 10 meters as a misfit?

388-389: "..Indeed, the extension and thickness of this contact is believed to be a driving element of groundwater dynamics in the area of the Colosseum." --> but "is believed to be" is different from it is, don't you? Please give more insight about.

404-408: “The model was then refined by adjusting the geometry of the Middle Pleistocene volcano sedimentary formations ... This operation was accomplished by iteratively changing the dip direction of geological contacts in the top surface of the model. However, this process generated unsatisfactory results, and the geological contacts ... were removed from the model input.”  I've to admit that this is a quite strong choice for me!!

410-411: "The release of surface topography constraints helped..". Unclear!!

425-427: "In particular, the logs provided geometric constraints (i.e. top and bottom surfaces, lateral extension) for the gravel bed at the base of the alluvium (SFTba 1 Complex)". Few data to provide constrains!

430-433: "Therefore, it can be hypothesised that the NW portion of the Colosseum foundation ring is anchored mainly in the Middle Pleistocene sandy terrains of the FTR 2 Complex, instead of being anchored to the Upper Pleistocene-Holocene alluvium of the Labicano stream, as considered in previous studies ([13], [14])", while it sound as a valuable information from your work,  seems to me that FTR2 Complex may be located in W-NW and not in NW (fig.5). S3 borehole log, at north, cross, show contact with CIL and not with FTR2.

438: "..a 1-2 m thick alteration zone was detected, which can be ascribed to paedogenetic processes acting.." Do you have more info about the alteration zone to show? Alteration may be not related to pedogenic processes.

549: fig 10. From this draw the Colosseum foundations are fully placed in the backfill deposits. Is it a mistake, right?

In summary, I find the job too long. The introductory part, for example, develops excessively as does chapter 2 (which is also very interesting). I suggest shortening if possible.
Rather, I believe that a role should be given, through a discussion, to anthropocene deposits, which as shown for example in figure 7 represent an element of importance (not only volumetric) and that in combination with other anthropic elements (ancient, recent and current ) can in some way modulate the effects of events such as that reported also in the text of 20 October 2011, which characterized the city of Rome with intense flash flood phenomena.